# Different brain networks mediate the effects of social and conditioned expectations on pain

Leonie Koban [1,2,3,4], Marieke Jepma[5], Marina López-Solà[6] & Tor D. Wager [1,2,7]

Information about others' experiences can strongly influence our own feelings and decisions. But how does such social information affect the neural generation of affective experience, and are the brain mechanisms involved distinct from those that mediate other types of expectation effects? Here, we used fMRI to dissociate the brain mediators of social influence and associative learning effects on pain. Participants viewed symbolic depictions of other participants' pain ratings (social information) and classically conditioned pain-predictive cues before experiencing painful heat. Social information and conditioned stimuli each had significant effects on pain ratings, and both effects were mediated by self-reported expectations. Yet, these effects were mediated by largely separable brain activity patterns, involving different large-scale functional networks. These results show that learned versus socially instructed expectations modulate pain via partially different mechanisms—a distinction that should be accounted for by theories of predictive coding and related top-down influences.

[1] Institute of Cognitive Science, University of Colorado Boulder, Muenzinger D244, 345 UCB, Boulder, CO 80302, USA. [2] Department of Psychology and Neuroscience, University of Colorado Boulder, Muenzinger D244, 345 UCB, Boulder, CO 80302, USA. [3] Brain and Spine Institute (ICM), Control-Interoception-Attention Team, 47 Boulevard de l'Hôpital, 75013 Paris, France. [4] Marketing Area, INSEAD, Boulevard de Constance, 77300 Fontainebleau, France. [5] Department of Psychology, University of Amsterdam, Nieuwe Achtergracht 129B, 1018 WS Amsterdam, The Netherlands. [6] Division of Behavioral Medicine and Clinical Psychology, Department of Pediatrics, Cincinnati Children's Hospital Medical Center, University of Cincinnati, 3333 Burnet Avenue, MLC2 7031 Pain Research Center, Cincinnati, OH 45229, USA. [7] Department of Psychological and Brain Sciences, Dartmouth College, HB 6207, Moore Hall, Hanover, NH 03755, USA. Correspondence and requests for materials should be addressed to L.K. (email: leonie.koban@colorado.edu)

Expectations and beliefs shape human experience and behavior in many important ways. They are thought to serve as priors—initial belief distributions—in Bayesian and predictive coding frameworks[1,2], shaping sensory perception via feedback projections[3] and guiding action and decision-making[2]. They are core features of persistent mindsets that shape clinical symptoms, physiology, and other life outcomes[4]. Prominent examples illustrating the power and complexity of expectations are placebo and nocebo effects—changes in subjective experience and objective physiology due in part to expectations about medical treatment and symptom relief[5–7].

While expectations and related forms of prior information are often treated as a unitary concept, they can be based on different sources. In many cases, expectations are based on what we have learned from our own prior experience, via classical conditioning or other forms of associative learning. In this case, prior expectations can help to stabilize the perception of otherwise noisy or ambiguous input[8,9]. However, expectations can also stem from secondary sources, such as what others tell us about their experiences. Vicarious experiences communicated via symbols or words—termed here "social information"—allow us to form priors about experiences we have never had. Several studies demonstrate that information from both vicarious and direct experiences influence feelings and decisions, and may even have synergistic effects on learning and behavior[10–16].

However, it remains unclear whether similar or different neural systems mediate direct experience-driven and vicarious influences. Experimental studies often create "top-down" effects via a combination of associative learning (i.e., classical conditioning) and social information (e.g., instructions by an experimenter or doctor), without trying to separate their sources[5–7,17–22]. Thus, the commonalities and differences in the neurophysiological pathways involved remain largely unexplored[15]. Here, we create expectations about pain based on two different sources, social information and associative learning, in order to dissociate their underlying brain mechanisms.

While prediction and predictive coding are thought to be central to information representation in multiple neural circuits throughout the brain[2,7,23], there is also evidence for specific systems representing expectations[24]. Key regions include the ventromedial prefrontal cortex (vmPFC)[25] and dorsolateral prefrontal cortex (dlPFC)[26,27], which are thought to have central roles in decision-making[28], instruction effects[16,29–32], and top-down biasing of information flow based on context[33,34]. Further, expectation effects on perceptual decision-making have been shown to involve the hippocampus[35], in line with a role for this region in both retrospective memory and prospective thought[36,37]. Similarly, placebo and nocebo effects, which typically combine both suggestions and reinforcement to induce expectations, are thought to operate via recruitment of dlPFC and vmPFC and their interactions with subcortical regions[6,38–41]. Yet, differential effects of social suggestions and learning have not been disentangled.

Here, we employed a recently developed experimental paradigm that allowed us to independently manipulate conditioned cue information and social information[42]. In each of 96 trials of a learning task, 38 participants were presented with two types of information (Fig. 1a). In order to investigate associative learning effects on pain, they were shown one of two conditioning stimuli (CS, drawings of an animal and vehicle), which were predictive of low-to-medium ($CS_{LOW}$, 50% 48 °C and 50% 49 °C) or medium-to-high ($CS_{HIGH}$, 50% 49 °C and 50% 50 °C) thermal pain stimulation. Independent of these cues, in order to investigate social instruction effects, we presented participants with the pain ratings of ten other fictive participants (Fig. 1a). This social information depicted either low ($Social_{LOW}$) or high pain ratings ($Social_{HIGH}$).

However, unbeknown to the participants, social cues were not predictive of actual thermal stimulation. Participants indicated how much pain they expected, then received a brief thermal stimulation, and rated experienced pain at the end of each trial (Fig. 1b).

We employed multilevel mediation to (1) characterize changes in pain-related brain activity for $Social_{HIGH} > Social_{LOW}$ trials and $CS_{HIGH} > CS_{LOW}$ trials and to (2) identify brain mediators of these experimental manipulations on pain ratings. We tested two competing predictions: if associative learning and social information effects have common brain mechanisms, brain effects should be found in similar regions. Alternatively, if the effects of conditioning and social information have separable underlying mechanisms, brain effects should be seen in distinct areas, such as "mentalizing" and prefrontal areas for social information effects and associative learning-related regions, such as amygdala and hippocampus for conditioned effects on pain.

Our results show that both types of effects are mediated by consciously reported expectations, but their brain mediators are largely different. Limbic and posterior brain regions mediate conditioning effects, while prefrontal and parietal regions more strongly mediate social information effects on pain. Together, these results suggest that top-down modulation of experience can stem from distributed sources in frontoparietal, limbic, and default mode areas, depending on the source of information. Expectations induced by social influence and instructions might be especially powerful in shaping perception and experience, by bypassing learning networks and directly engaging higher-level prefrontal and parietal association cortices.

## Results

**Behavior and skin conductance.** Higher stimulus temperatures (50 °C > 49 °C > 48 °C) significantly increased pain ratings ($\beta = 7.55$ [95% CI 5.88–9.22], $t(35) = 8.85$, $p < 0.001$, Cohen's $d = 1.50$). Consistent with our predictions, both social information and CS indicative of high (vs. low) pain increased ratings of pain expectations (Fig. 1c; $Social_{HIGH}$–$Social_{LOW}$: $\beta = 11.75$ [9.59, 13.92], $t(35) = 10.65$, $p < 0.001$, Cohen's $d = 1.80$; $CS_{HIGH}$–$CS_{LOW}$: $\beta = 1.45$ [0.51, 2.38], $t(35) = 3.02$, $p = 0.005$, Cohen's $d = 0.51$). The effects of social information on expectations were much stronger than those of the CS ($t(35) = 7.66$, $p < 0.001$, Cohen's $d = 1.28$). The Social effect on expectations decreased slightly over time (cue x time interaction, $\beta = -0.05$ [$-0.08$, $-0.03$], $t(35) = -4.57$, $p < 0.001$, Cohen's d = 0.77), but was present throughout the experiment (Fig. 1d). In contrast, CS effects on expectations only emerged over time, reflecting learning (cue x time interaction, $\beta = 0.04$ [0.01, 0.06], $t(35) = 3.02$, $p = 0.004$, Cohen's $d = 0.51$).

Both types of information strongly influenced pain as well (Fig. 1c): heat at the same, medium-level intensity (49 °C) was rated as more painful in the $Social_{HIGH}$ compared with the $Social_{LOW}$ condition ($\beta = 4.78$ [3.27, 6.29], $t(35) = 6.19$, $p < 0.001$, Cohen's $d = 1.05$), and in the $CS_{HIGH}$ compared with the $CS_{LOW}$ condition ($\beta = 0.74$ [0.28, 1.20], $t(35) = 3.13$, $p = 0.003$, Cohen's $d = 0.53$). In parallel to expectations, pain ratings were more strongly influenced by social information than CS cues ($t(35) = 4.86$, $p < 0.001$, Cohen's $d = 0.81$). While CS effects on pain became stronger with time (reflecting learned pain modulation, $\beta = 0.02$ [0.0007, 0.038], $t(35) = 2.03$, $p = 0.049$, Cohens' $d = 0.34$), the social influence on pain ratings remained stable over time (i.e., cue x time interaction: $p > 0.20$, Fig. 1e).

Expectations formally mediated both Social and CS effects on pain (path $ab$ effect for Social, $\beta = 5.56$ [4.31, 6.81], $t(35) = 8.56$, $p < 0.001$, Cohen's $d = 1.47$; for CS, $\beta = 0.47$ [0.18, 0.76], $t(35) =$

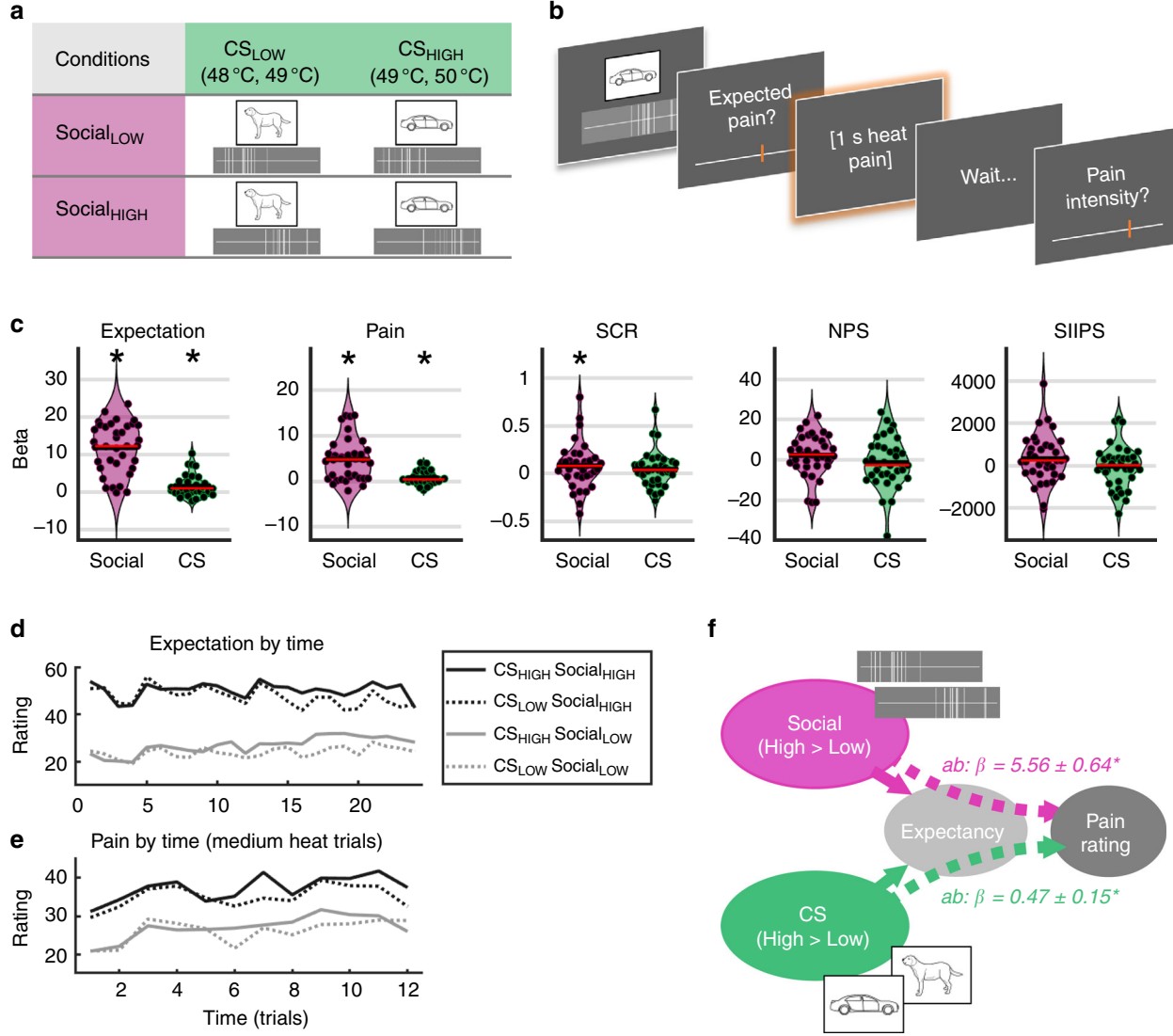

**Fig. 1** Design and results. **a** Design. The social information could be either low or high on average (Social$_{LOW}$ or Social$_{HIGH}$), but was not correlated with actual heat intensity. In contrast, the learning cues (CS$_{LOW}$ and CS$_{HIGH}$,) were followed by low-to-medium or medium-high heat intensity, respectively. **b** Each trial started with the simultaneous presentation of social information ("pain ratings of previous participants", depicted as vertical lines on a VAS) and one of the CS (animal or vehicle drawing). Participants then rated their pain expectation, received a short heat pain stimulation, and rated how much pain they experienced. **c** Effects on behavior, physiology, and brain patterns. Violinplots show the effects for Social$_{HIGH}$ > Social$_{LOW}$ and CS$_{HIGH}$ > CS$_{LOW}$. Each dot reflects the beta (effect magnitude) estimate of one participant. Both social information and CS significantly influenced expectation (social information: $t(35) = 10.65$, $p < 0.001$, CS: $t(35) = 3.02$, $p = 0.005$) and pain ratings (social: $t(35) = 6.19$, $p < 0.001$, CS: $t(35) = 3.13$, $p = 0.003$). Skin conductance responses (SCR) were significantly modulated by social information ($t(35) = 2.04$, $p = 0.049$), but not by CS. Neither NPS nor SIIPS showed significant responses to social information or CS. Asterisks denote significant effects at $p < 0.05$ (using $t$ tests). **d**, **e** Time course of expectation and pain ratings. The difference between dotted and solid lines reflects the CS effect, and the difference between gray and black lines the social information effect. CS effects on expectation and pain increased over time (interaction effects CS*Time on expectation: $t(35) = 3.02$, $p = 0.004$, on pain: $t(35) = 2.03$, $p = 0.049$). Social information effects on expectations and pain remained significant throughout the experiment, but decreased over time for expectation ratings (Social × Time on expectations: $t(35) = -4.57$, $p < 0.001$). The x-axis shows trials per condition. **f** Behavioral mediation analysis. Expectation ratings significantly mediated both Social ($t(35) = 8.56$, $p < 0.001$) and CS effects on pain ($t(35) = 3.27$, $p < 0.001$). Source data for panels **c**, **d**, and **e** are provided as a Source Data file

3.27, $p < 0.001$, Cohen's $d = 0.53$, Fig. 1f; see also Supplementary Fig. 1), replicating our previous findings[42].

Paralleling the behavioral findings, skin conductance responses (SCR) to pain (Fig. 1c; Supplementary Fig. 2) were higher in Social$_{HIGH}$ than Social$_{LOW}$ trials ($\beta = 0.08$ [0.003, 0.16], $t(35) = 2.04$, $p = 0.049$, Cohens' $d = 0.34$). SCR were only numerically higher for CS$_{HIGH}$ compared to CS$_{LOW}$ trials ($t(35) = 1.47$, $p = 0.15$). In contrast to our earlier findings, expectation ratings did

not significantly mediate social information and CS effects on SCR ($t$ test, both $p$'s > 0.20), potentially due to noisier physiological data in the MRI environment.

**Effects on established multivariate brain measures related to pain.** We next tested for effects on two previously described multivariate brain markers of pain. The neurological pain

signature (NPS[43]) was optimized and validated to predict pain ratings with high accuracy during evoked pain related to multiple types of noxious stimuli. It is thought to reflect primarily pain of nociceptive origin[43], as it does not respond to several forms of psychological pain modulation[44,45]. The stimulus-intensity independent pain signature (SIIPS1)[46] was developed to predict pain independent of nociceptive input and mediates effects on pain of several psychological manipulations, including cued expectancy and perceived control[46]. Here, while single-trial fMRI responses in both NPS and SIIPS were significantly associated with trial-by-trial pain ratings (NPS: $\beta = 0.032$ [0.015, 0.049], $t(35) = 3.69$, $p < 0.001$, Cohens' $d = 0.62$; SIIPS: $\beta = 0.0002$ [0.00008, 0.0004], $t(35) = 3.06$, $p = 0.004$, Cohens' $d = 0.52$), neither of the two patterns was significantly modulated by either the social information or the CS ($t$ tests, all $p$'s $> 0.1$, see Fig. 1c). This suggests that social information and learning effects on pain may not influence primary nociceptive processing, but exert their effects via other mechanisms.

**Whole brain mediation analysis.** We estimated two multilevel-mediation models[17,41,47] to assess the mechanisms of social information and associative learning on pain-related brain activity modeled on a single-trial level (see Fig. 2a). In brief, $Social_{HIGH}$ versus $Social_{LOW}$ and $CS_{HIGH}$ versus $CS_{LOW}$ served as predictors (coded 1 and −1 for high and low, respectively), and single-trial pain ratings served as the outcome. Trial-by-trial beta images for medium-temperature pain events (matched on stimulus intensity) were included as mediators. CS cues served as first-level covariates in the Social→Brain→Pain model, and vice versa for the CS→Brain→Pain model, so that both models controlled for both cue effects. Path $a$ effects describe changes in activity based on the manipulation (similar to a classic 2nd level contrast), i.e., $Social_{HIGH} > Social_{LOW}$ and $CS_{HIGH} > CS_{LOW}$. Path $b$ characterized brain activity that predicts pain ratings independent of the two experimental manipulations. Path $ab$ (the $a*b$ product or "mediation effect") characterized brain activity that significantly mediates the effect of the manipulations (social information and CS) on pain ratings. An extended overview of mediation is provided in Supplementary Fig. 1.

**Cue-independent contributions to pain.** We first identified brain areas associated with higher pain ratings independent of social information and CS cues (Path $b$, q < 0.05 FDR-corrected, yellow in Fig. 2b). These included areas typically involved in pain processing, such as the dorsal anterior cingulate cortex (ACC) and premotor cortex, mid and posterior insulae, somatosensory cortex (leg area, corresponding to the stimulation site), ventrolateral thalamus, and cerebellum, as well as occipital (visual) cortex.

In order to assess how these regions map onto large-scale networks, we calculated the spatial pattern similarity (Pearson correlation coefficients) of the mediation maps to seven cortical resting-state networks[48]. The wedge plot in Fig. 2b shows these similarity values, with the radius of each wedge proportional to the correlation strength and area proportional to variance explained. Pain-predictive activity independent of manipulations was concentrated in somatomotor, ventral attention, and visual networks (see Supplementary Tables 1, 6).

We further calculated the similarity between the Path $b$ map and association test $z$-score brain maps of 525 terms from the Neurosynth meta-analytic database[49]. While relationships with terms should be interpreted with caution and are only suggestive, they can be useful for contextualizing findings and provide a more fine-grained comparison with existing large-scale databases than the canonical seven-network parcellation. The top ten

unique terms associated with pain-related (path $b$) effects were themed around pain processing (e.g., "noxious"', "heat", "pain", "painrelated") and somatomotor function ("foot", "limb", "sensation", "somatosensory", "muscle", "voluntary", see Supplementary Table 7). Thus, meta-analysis-based decoding here seemed related to the stimulation to the right calf muscle.

**Social information effects on pain.** Increased activity for the $Social_{HIGH}$ compared to the $Social_{LOW}$ condition was found in several cortical and subcortical areas (path $a$ effects, purple in Fig. 2c, FDR q < 0.05 corrected, Supplementary Table 2). First, they comprised areas associated with cognitive control and top-down attention, including dlPFC, inferior parietal lobule (IPL), and intraparietal sulcus (IPS). Second, they included brain areas associated with salience, affect, and pain processing, i.e., ACC, anterior insula and operculum, and ventrolateral thalamus. Third, increased activity was found in brain areas involved in somatosensory integration, such as mid insula and parietal operculum. In addition, inferior temporal and visual cortex showed increased activation for $Social_{HIGH} > Social_{LOW}$.

These social information effects on pain were spatially correlated mostly with the dorsal attention and frontoparietal networks (see Supplementary Table 6). Paralleling the network findings, the top ten decoding results for $Social_{HIGH} > Social_{LOW}$ using Neurosynth meta-analytic maps included terms associated with attention and cognitive control (e.g., "topdown", "distractor") and with word and number processing (e.g., "counting", "orthographic", "number", "lexical", Supplementary Table 7). Thus, though some regions identified are reliably involved in social cognition, we do not have strong reasons to believe that the processes engaged are uniquely social; they may reflect effects of control and attention on pain more generally.

Mediators of social information effects on pain ratings (path $ab$, purple in Fig. 2d; Supplementary Table 3) included a similar set of regions, especially vlPFC, dorsomedial prefrontal cortex (dmPFC), dlPFC, IPL/IPS, visual cortex, inferior temporal sulcus, orbitofrontal cortex, and visual cortex. In close correspondence to path $a$ effects, the networks most prominently involved in mediation were the dorsal attention and frontoparietal networks, with a smaller contribution of the default mode network (see Supplementary Table 6). The top ten decoding terms based on Neurosynth included words associated with cognitive control and attention (e.g., "memory", "maintenance", "switch", "executive", "attentional", see Supplementary Table 7).

We next used a conjunction analysis to identify spatial overlap in brain areas responding to the $Social_{HIGH} > Social_{LOW}$ manipulation (path $a$) and those mediating the effects of this manipulation on pain outcomes (path $ab$). The mediation effect is driven by a combination of responses to the experimental manipulation, correlations with pain, and correlations in individual differences between these two effects (i.e., stronger responses to the $Social_{HIGH} > Social_{LOW}$ manipulation correlated with stronger effects of the brain region on pain). Thus, the $ab$ effect encompasses all the elements required to link the manipulation, brain, and behavior, but does not guarantee that all significant regions show significant cue effects. The conjunction analysis therefore shows regions that show both $Social_{HIGH} > Social_{LOW}$ effects and mediation effects (purple in Fig. 3a). It confirmed the important contribution of the dlPFC, IPS, dmPFC, vlPFC, and visual cortex to social information effects on pain. Higher single-trial activity averaged over this set of regions was correlated with higher single-trial expectation ratings ($\beta = 0.86$ [0.33, 1.40], $t(35) = 3.15$, $p = 0.003$, Cohen's $d = 0.53$), suggesting that these areas were involved in the generation of explicit expectations based on the social

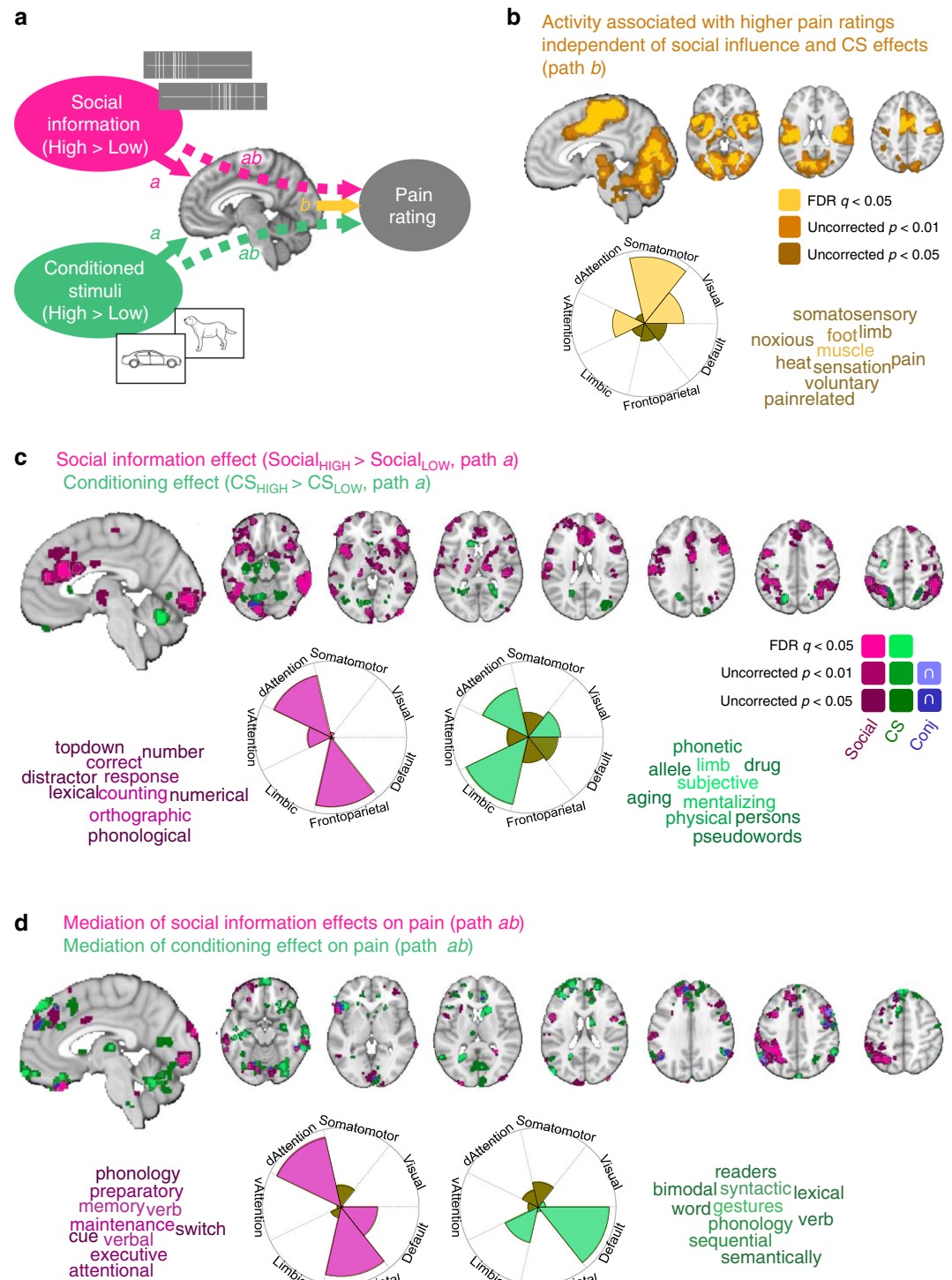

information. Individual differences in how much social information influenced expectation ratings did not moderate this relationship between brain activity and expectations ($t$ test, $p > 0.10$).

**Learning effects on pain.** Increased activity for $CS_{HIGH}$ compared with $CS_{LOW}$ was found in a largely different set of brain regions (path $a$, green in Fig. 2c), including "limbic" areas such, as hippocampus, caudate, and retrosplenial cortex, as well as cerebellum, precentral gyrus, and left IPS. These activations mapped

weakly onto a combination of limbic, dorsal attention and visual networks (Fig. 2c; Supplementary Tables 4, 6). Accordingly, the top ten Neurosynth associations for the $CS_{HIGH} > CS_{LOW}$ map spanned over various topics including language ("pseudowords", "phonetic") and mentalizing ("persons", "mentalizing") (see Supplementary Table 7).

Significant mediation of learning effects on pain (path $ab$) was seen in many areas, including dmPFC, medial and lateral orbitofrontal cortex, left anterior insula/operculum, caudate, hippocampus, retrosplenial cortex, fusiform gyrus, dlPFC, IPL, and cerebellum (green in Fig. 2d; Supplementary Table 5). Some

**Fig. 2** Whole brain mediation analysis. **a** Overview. A mass-univariate mediation analysis was performed to identify: (1) activity increases for high > low social information and CS (Path $a$), (2) activity associated with increased pain ratings when controlling for path $a$ effects (path $b$), and (3) activity formally mediating the effects of social information and CS on pain ratings (dashed arrows, path $ab$). **b** Path $b$ effects. Significant pain-related activity independent of the experimental manipulations was found in mid cingulate, posterior and mid insula, thalamus, cerebellum, and other regions. The wedge plot (with the radius of each wedge proportional to the correlation strength) indicates a high spatial correlation of this effect with the Somatomotor network (Pearson correlation coefficient $r = 0.28$, see Supplementary Table 6). The ten most strongly associated Neurosynth terms are shown on the right (decreasing brightness indicates order of associations, see Supplementary Table 7). **c** Path $a$ effects for Social (purple), CS effects (green), and their conjunction (blue). Social information effects (increased activity for $Social_{HIGH} > Social_{LOW}$) were found in ACC, anterior insula, dlPFC, and parietal areas. Those effects most strongly mapped on the frontoparietal ($r = 0.13$) and dorsal attention networks ($r = 0.12$, see wedge plot and Supplementary Table 6) and were associated with terms reflecting cognitive tasks (see Supplementary Table 7). CS effects ($CS_{HIGH} > CS_{LOW}$) were seen in limbic areas and cerebellum, and showed a more diffuse mapping on large-scale networks and meta-analytic terms. **d** Path $ab$ effects for Social (purple), CS effects (green), and their conjunction (blue). Social influence effects on pain mapped on the frontoparietal and dorsal attention networks (both $r$'s = 0.06) and on terms associated with cognitive control. CS effects mapped on the default mode network ($r = 0.06$) and were associated with terms related to semantic processing. All maps were thresholded at FDR $q < 0.05$ corrected for multiple comparisons across the whole brain (gray matter masked) with adjacent areas thresholded at $p < 0.01$ and $p < 0.05$ (uncorrected) for display

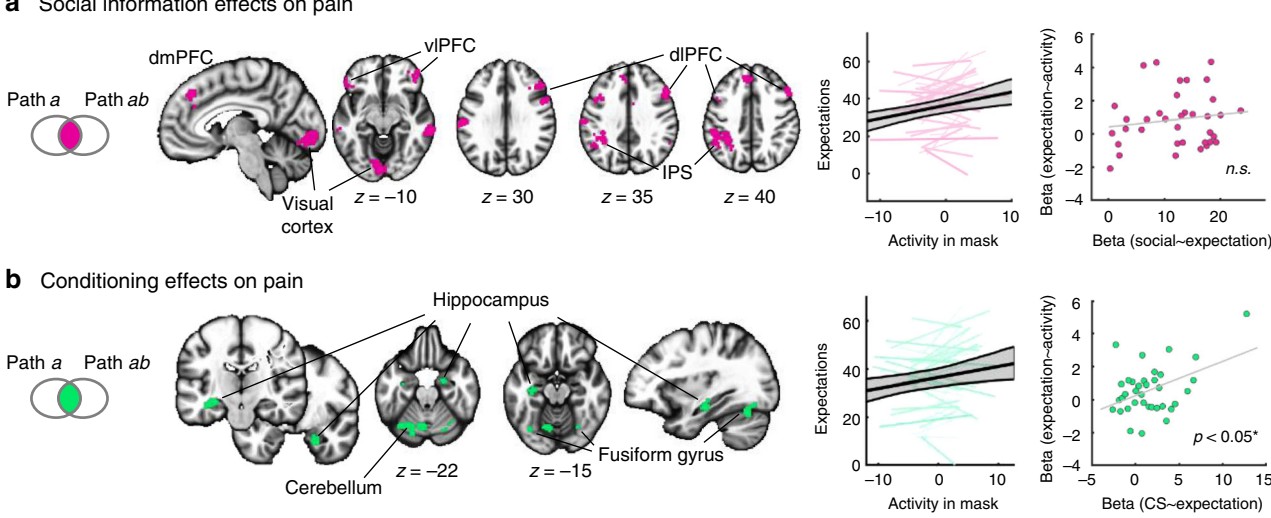

**Fig. 3** Conjunction of path $a$ and $ab$ effects. **a** Social information effects were consistent across paths $a$ and $ab$ in dmPFC, vlPFC, dlPFC, IPS, and visual cortex. Trial-wise average activity (betas) in these regions correlated significantly with trial-wise expectation ratings (individual slopes in purple). Average activity did not significantly predict individual differences in the social influence effects on expectations (scatter plot). **b** CS effects were consistent across paths $a$ and $ab$ in the hippocampus, cerebellum, and fusiform gyrus. Trial-wise average activity in these areas correlated significantly with trial-wise expectation ratings (individual slopes in green). Individual differences in the strength of the activation were correlated with individual differences in how much expectations ratings were driven by the CS. Conjunction effects are displayed as the intersection of activation for paths $a$ and $ab$, each of them thresholded at $P < 0.05$ FDR-corrected and adjacent voxels at $p < 0.05$ uncorrected. Asterisk reflects significant Pearson correlation coefficient ($p < 0.05$). Shaded error bands reflect bootstrapped 95% confidence intervals. Source data are provided as a Source Data file

of those effects partially overlapped or were neighboring with mediation effects of social influence. Yet, learning effects were mostly correlated with default and limbic networks and with meta-analytic maps associated with semantic processing (e.g., "lexical", "word", "semantically", "gestures", see Supplementary Tables 6, 7). As noted above, this does not strongly imply that these processes were involved, but suggests similarity to regions engaged in associative learning, including semantic associations.

We again used a conjunction analysis to illustrate the overlap in brain regions that showed increased activation to pain for $CS_{HIGH} > CS_{LOW}$ (path $a$ effects), and regions that statistically mediated the effects on pain ratings (path $ab$ effects). This conjunction revealed three main regions of bilateral effects: hippocampus, fusiform gyrus, and cerebellum (green in Fig. 3b). In parallel to the social mediator regions, higher activity in these areas again correlated with higher trial-by-trial expectation ratings ($\beta = 0.58$ [0.09, 1.06], $t(35) = 2.34$, $p = 0.025$, Cohen's $d = 0.39$; Fig. 3b, middle). This association was modulated by individual differences in how much expectations were driven by the learning cues ($\beta = 0.21$ [0.07, 0.35], $t(34) = 2.84$, $p = 0.008$,

Cohen's $d = 0.49$; Fig. 3b, right)—however, this relationship with individual differences was largely driven by a single participant (~3 STD above the mean on both brain and behavior). Larger sample sizes are needed to investigate individual differences in more detail. In sum, these results show the contribution of brain regions involved in memory, learning, and object recognition in learned pain modulation and (as with unconditioned social information) suggests a role for conscious (reportable) expectation in this process.

**Cue-related and interaction effects.** For completeness, we tested how social information and CS influenced brain activity at the time of cue presentation, again demonstrating largely different systems responding to each type of cue and mediating their effects on expectations (see Supplementary Fig. 3). We further explored interaction effects between social and CS conditions, shown in Supplementary Fig. 4. Broadly, these analyses confirmed the importance of frontoparietal, orbitofrontal, and temporal systems in pain expectancy, with ventral and dorsal systems

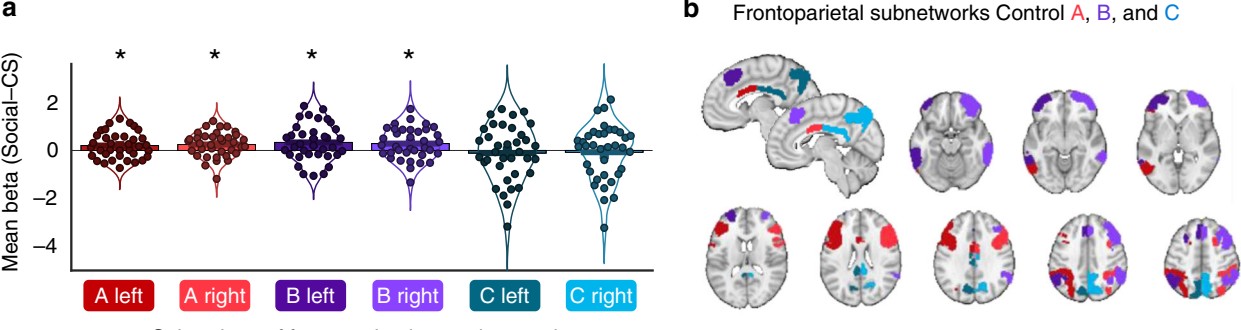

**Fig. 4** Difference between Social and CS effects in frontoparietal control subnetworks. **a** Difference in mean path *a* beta weight (Social–CS) in Control A (dark and bright red), Control B (dark and bright purple), and Control C (dark and bright blue) subnetworks in the left and right hemisphere (dark and bright colors, respectively). Each dot reflects the difference in mean beta estimates of one participant. Significantly greater activation for social information was found in the Control A (left: $t(35) = 2.3$, $p = 0.027$, 95% CI = [0.03, 0.39], Cohen's $d = 0.38$; right: $t(35) = 2.9$, $p = 0.006$, 95% CI = [0.08, 0.43], Cohen's $d = 0.48$) and the Control B (left: $t(35) = 2.6$, $p = 0.012$, 95% CI = [0.08, 0.59], Cohen's $d = 0.44$; right: $t(35) = 2.6$, $p = 0.013$, 95% CI = [0.07, 0.51], Cohen's $d = 0.44$), but not in the Control C network. Asterisks denote networks with significant differences between social information and CS effects (using t-tests, $p < 0.05$). Source data are provided as a Source Data file. **b** Display of frontoparietal control subnetworks A (dark and bright red), B (dark and bright purple), and C (dark and bright blue) on sagittal and transversal brain slices

preferentially involved in associative learning and social information effects, respectively.

**Similarity versus separability of social influence and learning effects**. To test for commonalities between social information and learning effects, we performed a conjunction analysis (using a conjunction null[50]). At a lenient threshold ($p < 0.05$ uncorrected voxels adjacent to FDR-significant voxels), CS and social influence showed a few small clusters of common path *a* effects in visual cortex and bilateral superior parietal lobule (see Fig. 2c). To quantify the degree of overlap vs. dissociation, we computed the Dice coefficient for voxels activated at 0.05 uncorrected. The Dice coefficient—which can theoretically range between 0 and 1, where 0 reflects complete separation and 1 reflects perfect overlap—for path *a* was 0.024, thus suggesting relatively little overlap between path *a* activations for Social and CS effects. For path *ab*, shared mediation effects of social influence and CS effects on pain were observed at $p < 0.01$ and $p < 0.05$ uncorrected, notably in dmPFC, dlPFC, vlPFC, and IPL (see Fig. 2d), with a Dice coefficient of 0.081. Thus, while these clusters of conjunction effects were relatively small in size, this suggests that parts of the frontoparietal network may be involved in top-down modulation of pain based on expectations irrespective of their sources. In sum, mediation maps suggest a dissociation in the regions mediating both learned and instructed cue effects on pain, along with some potential similarities in frontal and parietal regions.

While the peak areas involved in each type of pain modulation are distinct, it is conceivable that they reflect similar underlying activity patterns with distinct peaks surviving significance thresholds. In order to test this possibility, we performed several additional analyses.

First, we assessed whether social influence and CS-related activity could be separated based on distributed multivariate activity. We trained a support vector machine (SVM) classifier using leave-one-subject-out cross-validation to separate individual beta images for $Social_{HIGH} > Social_{LOW}$ from beta images for $CS_{HIGH} > CS_{LOW}$ (path *a* effects) in all brain areas associated with mediation (in the a and ab conjunction images) for either social or learning effects (i.e., union of regions displayed in Fig. 3a, b). This SVM yielded significant and moderately accurate predictions (forced-choice, mean ± SE = 72% ± 7.5%, binomial test, $P = 0.011$), indicating separable, reliably distinct patterns of activity

for social and learning effects during pain in these brain regions (though it does not rule out potential similarities as well).

Second, since conjunction effects were found in frontal and parietal regions, we tested how individual (unthresholded and normalized) beta images for Social and CS path *a* effects engaged more fine-grained parcellations of the frontoparietal network (Fig. 4), based on an established 16-network cortical parcellation[48,51] (Supplementary Fig. 5). Overall, the frontoparietal network (but no other network) was significantly more activated for Social compared to CS path *a* effects ($t(35) = 3.1$, $p = 0.0042$, Cohen's $d = 0.51$). Further, the frontoparietal subnetworks "Control A" and "Control B" in both left and right hemisphere were more activated for the Social compared with the CS path *a* effect (see Fig. 4a). In contrast, both left and right "Control C" subnetworks did not differ between Social and CS effects. This is consistent with the observation that strong Social effects were observed in the prefrontal, lateral parietal, and temporal parts of the frontoparietal network, but not in the medial parietal cortex or posterior cingulate cortex (which constitute the "Control C" subnetwork, see Fig. 4b).

Third, we analyzed the spatial covariation between the unthresholded weight maps for Social and CS mediation effects (path *ab*), and summarized the voxel-level covariation within each network. Here, voxels are the unit of analysis, and voxels within a network may have diverse functional relationships with CS and social information effects. In Fig. 5 and Supplementary Fig. 6, we plotted the joint distribution of both type of effects—i.e., the weight (effect magnitude) of each voxel for the Social mediation effect on the *x*- and for the CS mediation effects on the *y*-axis, separately for each network[52,53]. Effects in any given voxel could be positive, negative, or near-zero for each of the Social and CS mediation effects. This lends itself to classifying voxels within each network into eight equally-sized octants depending on the relative Social and CS effects. Voxels in Octants 1 and 3 were selectively related to positive mediation of CS and social influence respectively. Octants 5 and 7 showed selective negative effects of CS or Social cues, respectively. Voxels in Octants 2 and 6 show positive and negative mediation effects that are common and in the same direction for both cue types. Voxels in these octants drive positive spatial correlations across voxels within the network as a whole, indicating overlap. Finally, voxels in Octants 8 and 4 are those with positive CS effects, but negative Social cue effects, or vice versa. Voxels in these octants drive negative spatial

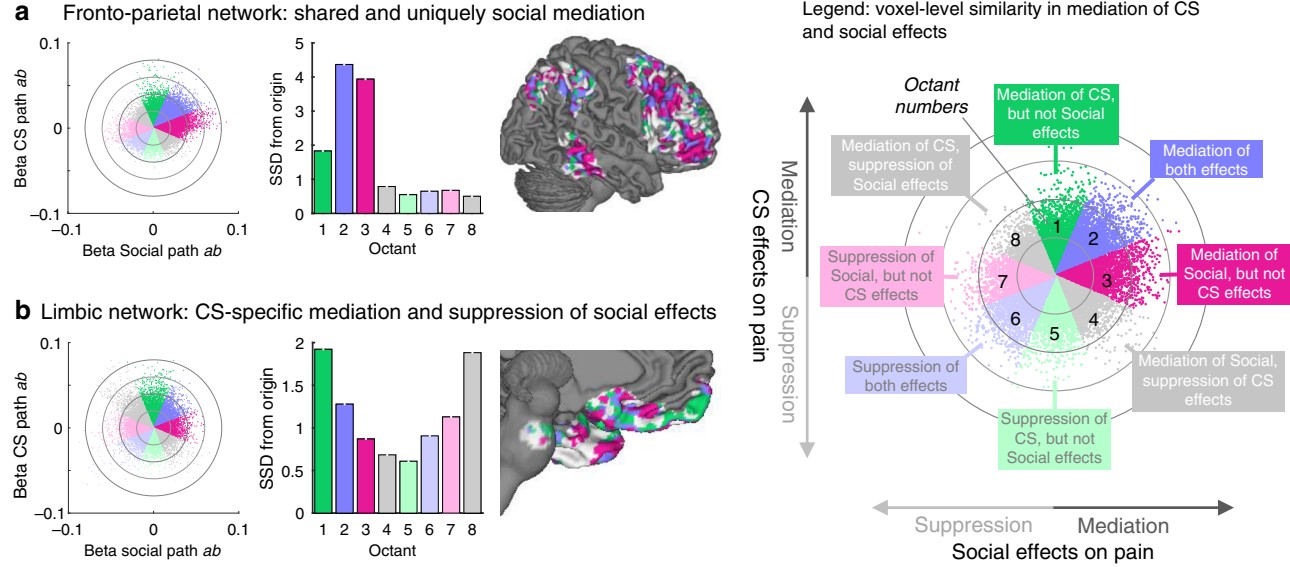

**Fig. 5** Voxel-level spatial covariation in Social and CS mediation (path *ab*) effects. Scatter plots display unthresholded single-voxel beta weights for CS (*y*-axis) and Social (*x*-axis) mediation effects (path *ab*). Each dot represents a voxel. Different colors are assigned to eight octants that reflect positive mediation for CS but not Social effects (Octant 1), positive mediation effects for both (Octant 2), positive mediation of Social, but not CS effects (Octant 3), and so on. Radial grids display distance from the origin (0,0) in 0.02 unit steps. These maps are descriptive, unthresholded illustrations of mediation beta weights. Units are arbitrary. **a** Voxel-level similarity in the frontoparietal network, showing the highest sum of squared distances (SSD, displayed in bar plots) from the origin in Octants 2 and 3, reflecting shared (CS and Social) positive and uniquely positive mediation of Social effects, respectively. Voxels in Octants 1–3 (green, blue, and purple, respectively) within the frontoparietal network (light gray area) are displayed on a lateral brain surface plot (far right). **b** Voxel-level similarity in the limbic network, showing highest SSD in Octants 1 and 8 (see bar plot), reflecting selective mediation of CS effects and suppression of Social effects. Suppression effects are in the opposite direction from, and thus "work against", the overall effects of cues on pain ratings. Voxels in Octants 1–3 (green, blue, and purple, respectively) in the limbic network (light gray area) are plotted on a medial brain surface (far right). See Supplementary Fig. 6 for the results in all seven large-scale networks. Source data are provided as a Source Data file

correlations, indicating dissimilarity. Furthermore, to provide an overall measure for voxels in each octant, we computed the sum of squared distances (SSD) from the origin, thus combining a measure of both absolute numbers of voxels in each octant and their (squared) distance from the origin.

This analysis of the spatial covariation of Social and CS effects shows qualitatively different patterns across networks. Visual, ventral attention, and default mode networks (Supplementary Fig. 6) have peak SSDs in Octant 2, reflecting a disproportionately large number of voxels that show positive (if not necessarily significant) effects for both Social and CS mediation, indicating some overlap in these networks. However, these networks also contained many voxels with positive weights only for CS (Octant 1) or only for Social mediation effects (Octant 3). Frontoparietal (Fig. 5) and dorsal attention networks show large effects in the shared positive (Octant 2) and in uniquely Social mediation effects (Octant 3). The limbic network (Fig. 5) shows a peak in voxels mediating CS, but not (or even suppressing) Social effects (Octant 1 and 8). Finally, the Somatomotor network was associated with many voxels showing suppressor effects for CS (negative slopes in mediation) and shared suppressor effects for CS and Social effects, in line with its role in primary nociceptive, but less in contextual pain modulation effects. Overall, this analysis provides more detailed evidence for shared and non-shared elements within each network. The networks with the strongest evidence for some shared processing include the dorsal and ventral attention, frontoparietal, and default mode networks, but these similarities are offset by the differential responses in the vast majority of voxels in these networks (the overall spatial covariance across voxels is relatively weak). Those with the strongest evidence for dissimilar effects of Social and CS

include the Limbic (CS preference) and Somatomotor networks.

## Discussion

The effects of both socially conveyed and learned expectations on human behavior and experience are pervasive. Dissecting the brain mechanisms of different sources of expectations—learned and socially instructed—is important for understanding a number of related and yet distinct phenomena such as placebo effects[5–7], social influence and instruction effects[22,54–56], the brain mechanisms underlying predictive coding[2,3,7], and effects of expectations and mindsets more broadly[4]. Here, we found that both social information and learning independently influenced expectations and pain reports, with especially large and durable (nonextinguishing) effects of social information. Future studies should test whether the presence of salient social information can augment or hinder experience-based learning, by comparing CS effects in the presence versus absence of salient social cues. The large effects of social information have important implications for decision-making in real life, where information from social media or online ratings often competes with experience-based learning of factual contingencies.

On a behavioral level, expectations mediated both types of effects—yet the brain effects and mediators were found in largely different brain regions and mapped onto different large-scale networks, suggesting dissociable effects of social information and learning on pain. These findings have implications for predictive coding theories of pain and of information processing more broadly[2]. Whereas previous studies have established the effects of expectations in many domains, it is less clear where these expectations come from and which brain systems are at the "top"

of "top-down" modulation of information processing[57]. Our results suggest a distributed system of source-dependent (e.g., social or conditioned cues) and source-independent brain areas involved in representing expectations and modulating pain accordingly. Current predictive coding models focus on the functional form of the models (e.g., hierarchical Bayesian information), but not on potential differences across priors based on sources and context. We show that priors in predictive coding frameworks may arise from different neural sources, even if they have common psychological correlates.

Mediators of social information effects on pain centered on frontoparietal regions frequently activated during cognitive control and attention tasks, whereas mediators of conditioned cue effects centered on the hippocampal, inferior occipitotemporal, and cerebellar circuits frequently identified in studies of memory and conditioning. This suggests that dlPFC and parietal activations often associated with placebo effects[5,6,41,58] may reflect direct effects of instructions and suggestions, whereas limbic or orbitofrontal activations may reflect learning effects, i.e., integration of experiences over time. The two systems also interact, and we found indications of common mediation effects, including in vmPFC, dmPFC, frontoparietal, and visual areas, suggesting potential common processes related to expectations or attention allocation to pain. Both types of pain modulation and their brain mediators were correlated with reported pain expectancies, suggesting that the differences across them are more related to the type of information presented and cognitive inference performed, rather than the engagement of conscious vs. unconscious predictions.

In addition, both types of effects differed to a large degree from activity that predicted pain independent of our experimental manipulations (Fig. 2b), which engaged primary nociceptive and sensory pathways. This suggests a direct influence of context and cognitive inference on reported pain experience, largely independent of influences on ascending nociceptive input. Such findings are consistent with a number of prior studies demonstrating similar direct effects of perceived control[46,59], reappraisal[60], and placebo[41,44].

Our results resonate with a large body of literature that has demonstrated the power of social context in shaping behavior, health, and affect[22,61,62], yet they are among the first to show how abstract social information can dramatically alter pain reports and physiological responses to pain[42,63]—even in the presence of predictive learning cues. Interestingly, while social information effects were seen in some areas associated with mentalizing, such as TPJ, dmPFC and adjacent ACC, other areas often associated with mentalizing (e.g., precuneus) were not key mediators of social information effects on pain. Instead, social information exerted its effects mostly via the recruitment of areas of the frontoparietal and the dorsal attention network such as dlPFC and IPS. These brain regions (especially dmPFC, vlPFC, and dlPFC) have been shown to play a role in social conformity effects on decision-making and value[55,64,65] and in instruction effects on aversive[16,31] and appetitive learning[29,30]. For instance, dmPFC responds to social conflict[55] and social dissonance[64], which lead to adjustments in decision-making and preferences. DmPFC has been associated with individual differences in confirmation biases that prevent extinction of instructed learning[66] and placebo-induced suppression of prediction errors in the ventral striatum[31]. Slightly more ventral areas in rACC have been proposed to be important for social learning and tracking other people's motivational states[67]. Further, learning by instructions is thought to be mediated by frontal systems, especially dlPFC[68]. Interactions of dlPFC with value-related and learning-related brain regions such as vmPFC, amygdala, and striatum may underlie instruction effects on associative learning[16,29,31]. dlPFC and IPS

have also been shown to play a role in the modality-independent representation of meaning and expectations[69]. Thus, though social influence did not have significant effects on primary targets of nociceptive afferents or validated brain predictors of nociceptive pain, they do alter circuits important for expectations, meaning, and choice. This study also shows that these circuits are directly linked to trial-by-trial pain experience.

What are the functional mechanisms underlying these effects? Comparisons with existing meta-analytic maps[49] and large-scale functional brain networks[48] suggest a primary mapping of social influence effects on pain to frontoparietal and dorsal attention networks associated with cognitive control, attention, and multiple forms of top-down influence[33,34,70]. These networks may influence representations in sensory integration areas, such as mid- and anterior insula. In other words, these systems may provide top-down attentional filters to align experience with socially instructed expectations. This interpretation is in line with a recent model of social information effects that suggests that dlPFC represents an "instructed state" that biases affective processing across other brain regions[22]. While frontoparietal and dorsal attention networks are not primarily thought of as "social cognition" or "mentalizing" regions, it is noteworthy that their functions (e.g., sustained attention and cognitive control) are thought to develop in a social context[71,72], and are associated with social effects on behavior such as social facilitation[73].

Contrary to intuition, associative learning effects were—while significant and meaningfully large ($d = \sim 0.5$)—much smaller in size than social information effects on pain. Despite strong effects of the different temperatures associated with high and low cues on pain ratings, not all participants learned the contingency between cues and pain (cf. Figure 1). In line with our previous findings[42], self-reported expectations mediated CS effects on pain. This reflects an important role of contingency awareness in this type of learned pain modulation, which is based on trace conditioning[74,75] (i.e., involving a delay between CS and reinforcer). In contrast, paradigms that use delay conditioning can elicit subliminal learning effects on pain[76,77].

Brain mediators of CS effects on pain included bilateral hippocampus, inferior occipital cortex, and cerebellum—regions distinct from those mediating social influence effects and those involved in basic nociceptive processing. Cerebellum and hippocampus are important structures for learning and conditioning. While the cerebellum is critically involved in both delay and trace conditioning[78,79], the hippocampus is necessary for trace conditioning, which requires contingency awareness[80,81]. Given its role in explicit learning and trace conditioning, the hippocampus is well positioned to associate painful stimuli with learned contingencies. While not a classical part of the pain-modulating network[82], the hippocampus is involved in pain processing across different studies[83–85], and our findings confirm previous evidence of hippocampal involvement in conditioning effects on pain[77,86] and in pain modulation more broadly[46]. Structural and functional alterations of the hippocampus are also a hallmark of chronic pain conditions[87–90], suggesting potential bidirectional effects between learning-circuits and pain.

In some respects, the distinction between social influence and conditioning effects shown here parallels other work on top-down versus bottom-up information effects[33,91]. Yet, there were also commonalities and overlapping brain effects. First, self-reported expectations mediated both effects. Second, activation in both sets of brain mediators was correlated with expectation ratings, demonstrating a presumed link with conscious awareness. Third, while there was no strong overlap in regions activated by each manipulation (path *a* effects), several networks (including default, frontoparietal, ventral attention, and visual) showed a large number of subthreshold voxels that showed positive mediation

effects for both experimental manipulations. Finally, a shared theme in the associated Neurosynth decoding results were terms associated with language and meaning (e.g., "verb", "phonology", "gestures"), again in line with the idea that conceptual processing may contribute to both learning and social influence effects on pain. Together, these results suggest that direct learning shapes expectations and pain via an interplay of limbic learning areas with frontal and parietal associations areas, whereas social instruction may engage some these circuits, particularly fronto-parietal control networks, more directly.

Social influence, instructions, and different types of learning constitute a large space of manipulations and effects. Here, we contrasted two specific manipulations that differed in several ways; matching them on all sensory and cognitive aspects is likely impossible and would reduce ecological validity. Whereas learning cues were reinforced using classical trace conditioning, the social information was not reinforced, i.e., not predictive of actual stimulus temperature. The learning cues were two drawings, whereas the social information was presented as ratings on a visual analog scale. While it is unlikely that those aspects caused the observed effects, future studies may use different cue types (e.g., numerical instead of visual representations of others' ratings) and other modalities of cue presentation[69] to replicate the present findings.

Further, in order to avoid a blurring of cue-evoked activity into the pain-related brain activity, pain stimuli were presented several seconds (jittered) after the cues, similar to previous studies[17,42,75,92]. This delay induces trace conditioning, which involves partially different mechanisms than "delay conditioning", which has been used in some other pain learning studies[21,77,86,93]. Different delays between cue and pain stimuli may involve different neurophysiological mechanisms. Future studies could also test whether trace and delay conditioning interact in different ways with social information and instructions. For instance, social information effects may be stronger or weaker when participants see the CS cues at the same time. This possibility is unlikely, however, since several studies have shown robust effects of social information on affect ratings and brain responses in the absence of conditioning cues[63,94].

We chose abstract social information (ratings), that are similar to online ratings or verbal information provided by doctors about other patients' experiences. The advantage of such abstract information is that its effects should be driven by purely conceptual processes without any intrinsic affective value (i.e., they are not "biologically prepared"[95]). Other types of social learning effects on pain[96–99] often involve more direct social interactions, including facial expressions or body language, which could be investigated in future studies. Such observational learning may be more similar to direct learning[100–102].

Further, we tested the similarity of our brain results with a 7-network parcellation of resting-state activity[48], and complemented this approach by investigating the contribution of subnetworks in each hemisphere and with analyses of the distribution of CS and social effects across individual voxels. Whether there is an optimal parcellation and/or degree of granularity for these pain-modulatory effects is an open question.

Finally, we tested effects on ratings of pain intensity, but not pain affect (unpleasantness). These two measures are often highly correlated[58,103]. Yet, they can be differentially modulated by contextual manipulations[104,105] and may correlate with different brain areas[106,107]. Future studies could test how social information and learning influence pain affect—a measure that is often more sensitive to social and contextual manipulations of pain[104].

## Methods

**Participants**. Thirty-eight healthy volunteers with no history of psychiatric, neurological, or pain disorders participated in the experiment. The data from two participants were excluded from all analyses due to large movements artifacts and delays in the experiment, leaving a final sample of 36 participants (20 female and 16 male, mean age = 27.1 years, range 18–50 years). All participants provided informed consent, and were paid for their time. The study was conducted according to the Declaration of Helsinki and approved by the Institutional Review Board of the Department of Psychology and Neuroscience at the University of Colorado Boulder.

**Materials and procedures**. At the beginning of each of 96 trials, participants saw two types of cues (see Fig. 1): Simple visual displays of putatively social information indicated the pain ratings of ten fictitious previous participants. These vicarious pain ratings were either low (Social$_{LOW}$) or high (Social$_{HIGH}$) on average, but were actually completely nonpredictive of subsequent heat stimulation, i.e., both Social$_{LOW}$ and Social$_{HIGH}$ cues were followed equally often by low (48 °C), medium (49 °C), or high (50 °C) heat pain stimulation. At the same time, participants were presented with one of two conditioning stimuli (CS, drawings of an animal or vehicle, see Fig. 1), which were partially reinforced with moderately painful heat (CS$_{LOW}$, followed by 48 °C or 49 °C stimulation) and more intensely painful heat (CS$_{HIGH}$, followed by 49 °C or 50 °C heat stimulation). Thus, learning cues (CS) were predictive of subsequent heat pain intensity and participants had to learn which CS predicted lower or higher pain on average.

For the social information, we generated 96 different stimuli (48 Social$_{LOW}$, 48 Social$_{HIGH}$), each depicting ten vertical white lines on a horizontal line with gray background that closely resembled the pain rating visual analog scale[42,63]. To generate a variable and plausible set of vicarious pain ratings, each stimulus was generated by sampling ten random values, restricted to be between 0 and 1, from one of two Gaussian distributions: $N(0.3, 0.15)$ for Social$_{LOW}$ and $N(0.7, 0.15)$ for Social$_{HIGH}$. For the CS, each participant was assigned one drawing of an animal (either a dog, cow, or horse) and one drawing of a vehicle (either a car, truck, or train) as the CS$_{LOW}$ and CS$_{HIGH}$, respectively (fully counterbalanced across participants).

Participants were instructed that we were interested in their subjective experience of pain, and how they would predict pain intensity by seeing pain ratings from previous participants and by learning about additional visual cues (animal or vehicle cartoons). They were told that one of the CS (animal or vehicle) was associated with higher pain on average, and the other cue with lower pain on average, and that their task was to learn these associations. Further, they were instructed that they would see the pain ratings of several other participants, presented as vertical lines on the same scale they would use to make their pain ratings. They then performed six runs (16 trials each) of the pain-learning task, corresponding to six nonoverlapping skin sites on the right calf chosen in randomized order. In order to habituate the participants to the stimulation on a new skin site[108], each run was preceded by a 49 °C stimulus (1 s plateau) before the scanner started again[42].

Each trial of the learning task started with the presentation (3 s) of one social information cue and one CS on a gray background (top or bottom position of social and CS cues counterbalanced across trials). After a jittered interval (2–4 s), participants used a visual analog scale (VAS, ranging from 0 to 100, anchored at "no pain at all" to "worst pain") to rate how much pain they expected (5 s). Then, after a jittered gray screen (2–4 s), they received a short (~2 s, 1 s plateau) heat pain stimulation of low (48 °C, 25% of trials), medium (49 °C, 50% of trials), or high (50 °C, 25% of trials) intensity to their right leg (see Fig. 1), using a 27 -mm diameter fMRI compatible CHEPS thermode controlled by a Medoc Pathway system (Medoc, Israel). Following another 4–8 s jittered interval, they were asked to rate how much pain they actually felt (pain rating, 5 s), using the same VAS as before. Between trials, participants saw a small white fixation cross on the same gray background (3–8 s). After the learning task, participants performed four runs of a generalization task, which will be reported elsewhere.

**Psychophysiological measures**. Electrodermal (skin conductance) activity was measured at the middle and ring finger of the (non-dominant) left hand. Pulse rate (not analyzed here) was measured at the left index finger. Both signals were recorded using a BIOPAC MP150 system and Acknowledge software at a sampling rate of 2000Hz. The data were low-pass filtered offline with a cutoff of 1 Hz.

**fMRI data acquisition and preprocessing**. Functional brain activity was acquired using a Siemens TrioTim 3T scanner, covering the brain in 26 interleaved transversal slices (3.4 -mm isotropic voxels), with a T2* weighted EPI GRAPPA sequence (TR = 1.3, TE = 25 ms, flip angle = 50°, FOV = 220 mm). Prior to preprocessing of functional data, time points that are potential global outliers (spikes) were identified based on meeting any of the following criteria: (a) absolute value of global signal > 10 median absolute deviations (m.a.d.), or (b) mahalanobis distance across slice-specific global means and spatial standard deviation > 10 median absolute deviations. These time points are identified on a per-run basis using recursive exclusion of outliers in a step-down test, so that outliers are removed before recursively identifying additional outliers (three iterations). SPM8 was used

for preprocessing for functional images, using a standard pipeline of motion correction, slice-time correction, spatial normalization to MNI space, and spatial smoothing of images using an 8-mm FWHM Gaussian kernel. For spatial normalization, T1-structural MPRAGE images (1 mm isomorphic voxels) were first coregistered to the mean functional image and then normalized to the SPM template using unified segmentation. Preprocessed functional images were resampled at a voxel size of $3 \times 3 \times 3$ mm.

**Analysis**. We used multilevel general linear models (GLM) to assess the effects of experimental conditions on single-trial expectation and pain ratings. Temperature effects (48, 49, 50 °C, coded as −1, 0, and 1) on pain ratings were assessed across all trials. Effects of CS and social information (both coded as −1 and 1, for low and high, respectively) on pain were assessed only in 49 °C trials (48 out of 96 trials), given that cues were not independent of stimulus temperature in 48 and 50 °C trials. Individual differences in Social and CS effects on expectations (first-level beta weights) were forwarded to correlations with average brain activity in mediator regions (see conjunction analysis). Multilevel-mediation analysis tested whether expectation ratings mediated Social and CS effects on pain. All statistical tests used a significance threshold of $p < 0.05$ (two-sided), and tests were planned contrasts with paired $t$ tests unless otherwise specified. Code for all analyses is available at https://github.com/canlab.

Single-trial SCR estimates were obtained using the SCRAlyze toolbox[109], which uses a GLM approach similar to standard event-related fMRI analysis. Each participant's skin conductance time series was low-pass filtered (1 Hz cutoff) and normalized. Regressors for single-trial pain onsets were convolved with a canonical SCR function[110] and fitted to the time series data, yielding single-trial beta estimates for SCRs. Parallel to the behavioral data analysis, multilevel GLMs were employed to test the effects of experimental conditions on SCRs in 49 °C trials.

Each participant's functional imaging data was submitted to a GLM that contained single-trials regressors for each heat pain stimulus (modeled as a stick function). The GLM further included event-related regressors for the four different cue conditions (2 Social × 2 CS), expectations ratings, and pain ratings (boxcar regressors with the duration of the response time). A total of 24 movement regressors (movement estimates for displacement and rotation in three dimensions, their derivatives, squared movement estimates, and derivatives of squared movement estimates) and spike regressors (coded as 1 for the outlier time point and zero for all other time points) were added as regressors of no interest to control for motion artifacts. Single-trial regressors that had variance inflation factors > 2.5 (indicating potential multicollinearity) were excluded from subsequent analyses. SPM8 and Matlab2014b were used for parameter estimation.

Single-trial beta estimates of pain-evoked functional BOLD activity in 49 °C trials (to control for temperature) were submitted to multilevel brain mediation analysis (code available at https://github.com/canlab). Brain mediation analysis identifies three statistical paths to characterize the causal effects of experimental variables (here: CS and social information) on outcomes measures (pain): (1) path $a$ characterizes the effect of the experimental factor ($CS_{HIGH} > CS_{LOW}$ and $Social_{HIGH} > Social_{LOW}$) on brain activity in each voxel, (2) path $b$ reflects the relationship between brain activity and the outcome, i.e., pain ratings, and (3) path $ab$ reflects brain activity formally mediating the causal link between experimental factor and brain outcome, thus significantly reducing the strength of the direct path $c'$. Resulting statistical maps were thresholded at $P < 0.05$ FDR-corrected across the whole brain and mediation paths[17,41] (corresponding to a voxel-level of $p = 0.0014$ for Social and $p = 0.0010$ for CS effects). Adjacent activity was thresholded at $p < 0.01$ and $p < 0.05$ uncorrected for displaying purposes. For completeness, additional mediation analyses were performed to examine the brain activity during the presentation of the cues (Social and CS) and their effects on expectation ratings. For this purpose, single-trial beta estimates of cue-evoked functional BOLD activity across all trials were used for two further multilevel mediation models (otherwise paralleling the models described above). The results of this analysis are shown in the Supplementary Fig. 5) at uncorrected threshold of $p < 0.001$ and adjacent activity at $p < 0.01$ and $p < 0.05$.

To illustrate the overlap of the brain effects with large-scale brain networks, we computed the spatial correlation (Pearson's $r$) of the thresholded maps with seven previously published resting-state networks as identified in a large ($N = 1000$ participants) sample[48]. Further, thresholded maps were compared with the Neurosynth database (525 term-based meta-analytic images, 2013 data release from neurosynth.org, Yarkoni, et al.[49]) to identify the ten unique most positively correlated terms associated with the activation maps. The Matlab text mining toolbox was used to create wordcloud plots of the decoded terms, with color brightness reflecting increasing correlation strength of terms with maps (arbitrarily scaled). To investigate the contribution of different parts of the frontoparietal network, we first normalized (z-scored) individual (unthresholded) path $a$ beta images and then computed the average activation in the frontoparietal network and in its three subnetworks on each side (right and left Control A, B, and C) for each subject, using a 17-network parcellation[51].

To test for overlap between social Information and CS effects, we computed the intersection of brain areas that showed effects for both type of cues at $P < 0.05$ FDR-corrected and adjacent voxels at $p < 0.05$ uncorrected, for both path $a$ and path $ab$ effects (Fig. 2). Further, to test for brain areas that showed both mediation (path $ab$) and path $a$ effects, we computed the intersection of brain areas that

showed both path $a$ and path $ab$ effects at $P < 0.05$ FDR-corrected and adjacent voxels at $p < 0.05$ uncorrected, separately for social Information (Fig. 3a) and CS effects (Fig. 3b). This test uses the conjunction null[49], which requires that each effect be significant. We visualized results from the conjunction analysis that were significant at an uncorrected level, but only if those were adjacent to voxels exhibiting effects at the more stringent FDR-corrected level. To test for the role of expectations within and between participants, we used spatially averaged beta estimates within these conjunction regions for each trial and regressed them to single-trial expectation ratings (using multilevel GLM, see slope plots in Fig. 3).

In order to illustrate the spatial similarity of Social and CS mediation effects in each large-scale network, we plotted each voxel's beta weight for the CS mediation effect (on the $y$-axis) as a function of the Social mediation effect (on the $x$-axis). We then divided the resulting scatter plot in eight octants that coarsely define directions of shared positive (Octant 2), shared negative (Octant 6), and uniquely Social or CS-positive and -negative unthresholded activations (unique Social positive in Octant 3, negative in Octant 7; unique CS positive in Octant 1, negative in Octant 5). Negatively correlated effects of Social and CS mediation would be seen in Octants 4 and 1. For illustration purposes, patterns of unique and shared effects, voxels falling in Octants 1 (uniquely CS mediation), 2 (shared), and 3 (uniquely Social mediation), were then plotted on lateral and medial brain surfaces (unthresholded).

**Reporting summary**. Further information on research design is available in the Nature Research Reporting Summary linked to this article.

## Data availability
The source data underlying Figs. 1c-e, 3a-b, 4a, 5a-b, and Supplementary Figs. 2a-c, 5, and 6b are provided as a Source Data file. Contrast images are available on Neurovault (https://identifiers.org/neurovault.collection:5677). Other data can be obtained from the authors upon reasonable request.

## Code availability
Code is available at https://github.com/canlab.

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

## Acknowledgements

We thank Megan Powell, Marianne Reddan, and Choong-Wan Woo for help with data acquisition. This work was funded by grants from the NIMH (R01 MH076136) and a VENI grant of the Netherlands Organization for Scientific Research (to M.J.). Matlab code for analyses is available at: https://github.com/canlab.

## Author contributions

L.K., M.J. and T.D.W. conceptualized the experiment. L.K. acquired and analyzed the data. T.D.W. obtained funding and supervised the project. L.K. wrote the original draft and created the figures. M.L.S., M.J. and T.D.W. contributed to the interpretation of the data and to the editing and revising of the paper.

## Additional information

**Competing interests:** The authors declare no competing interests.

