## [Transparent Peer Review File · Nature Communications]

Reviewers' comments:

Reviewer #1 (Remarks to the Author):

The present study investigates how social influences and learning effects on pain are mediated in the human brain. 36 participants performed an fMRI experiment in which they provided pain single-trial pain expectation and pain intensity ratings. Prior to the pain stimuli fictive pain ratings of 10 other participants and a visual conditioning stimulus were presented. Mediation analysis was performed to identify brain areas mediating the social influence and conditioning effects on pain. The results showed social and conditioning effects on pain expectation and pain intensity ratings with the social effects being much stronger than the conditioning effects. The results further show that the brain mediators of both effects fundamentally differed. Social influences were mostly mediated by brain networks related to attention whereas conditioning effects were mostly mediated by the default mode network.

Understanding different contextual modulations of emotions and pain is a highly relevant and timely topic. The methods of the study are sound and timely. The results are novel and convincing. The interpretation is adequate. A few explanations and clarifications might help to further improve this excellent manuscript.

1. The study includes a conjunction analysis of path a and path ab effects. This approach is novel and the underlying logic and its interpretation should be explained in plain terms.
2. The participants provide pain intensity ratings. Other studies have shown that pain intensity and pain affect can dissociate. The authors might discuss the distinction between pain intensity and pain affect, how the social and learning effects might have influenced pain intensity and pain affect and how possible differences in the effects on pain intensity and pain affect might account for differences in the underlying brain mediators.
3. The general reader is likely interested in the number of participants and trials which should already been included in the results section.
4. The figure legends should include more information on the underlying statistics.

Reviewer #2 (Remarks to the Author):

The study by Koban and colleagues takes an innovative approach to dissecting the effects of different sources of pain-altering expectations. The topic itself is of great interest and the authors address it from multiple angles: analysis of self-report and peripheral physiological data, mediation analysis of aforementioned data and fMRI data, correlation with large-scale brain networks and relating their effects to meta-analytic data. The manuscript is very well written and mostly easy to follow, though a few parts could be clarified. The sample size is adequate and is furthermore based on a behavioural study using a very similar paradigm. My main criticism (as detailed below) relates to the authors' analytical strategy and the robustness / reproducibility / generalizability of the 'social cue' vs 'conditioned cue' effects due to peculiarities of the paradigm.

1. According to the last paragraph of the introduction, the authors' main hypotheses are: "If associative learning and social information effects have common brain mechanisms, brain effects should be found in the same or similar brain regions. Alternatively, if the effects of conditioning and social information have separable underlying mechanisms, brain effects should be seen in distinct brain areas". However, I struggle to find a formal statistical test in the paper that tests for commonalities or compares the evidence for one versus the other hypothesis. Commonalities could for example be investigated with a simple conjunction analysis (for path a, path ab or a combination of the two). Their SVM analysis (L299) does not really address this point, as it doesn't exclude the possibility of shared activations (as the authors state themselves), but only shows that differentiation is possible. Could they maybe also test how much variance is shared between SOCIAL and CS effects in terms of loading on different large-scale networks (their Figure 2)?

2. With regards to the effects of social and conditioned cues, it seems to me as if the paradigm design might prevent any strong effects of conditioned cues on pain. First, there is a 50% reinforcement ratio, which is quite low and will lead to slow acquisition of contingencies (which is compounded by presentation of the social cues at the same time). Second, the choice of using trace conditioning will also play a role, as the delay between cue and pain is longer than the interval between the end of the current trial and the start of the next – such timing is known to weaken associative learning. At the same time, the peculiarities of the design might actually boost the social effects: since participants were explicitly instructed to detect the CS-US contingencies, their attention will have been focussed on the CS cues and one might wonder whether they had enough cognitive capacity to detect to non-informativeness of the social cues with regards to the level of pain. So I am not sure the large social effect will be easily replicated in other experiments once certain parameters of the design are changed.

3. I am also not sure whether the differential mapping on large-scale networks will be a robust phenomenon. For example, the way the social information was presented might automatically involve fronto-parietal systems due to the complexity of the input and the need to average the ratings to form a mean (this might be replayed during the pain stimulation). Would one expect to see a similar effect if the social information had instead been presented with strongly reduced complexity, e.g. as one number reflecting the mean rating, which would need much fewer cognitive resources? I also wonder about the level of granularity chosen here for the large-scale networks: is seven networks the optimal level for looking at such specific effects? Might it not be helpful instead to investigate this (very interesting) question across different levels of granularity in order to check whether similar effects of separable networks are observed? In my eyes, this would clearly strengthen the paper.

4. There are also some analyses that could add valuable information to the paper. The authors aim to dissociate the effects of social cues and conditioned cues on pain, but they do not report any data on the effects the cues had on peripheral physiological and brain data at the time of cue presentation. I would urge the authors to do so, as the cue response might be interesting with regards to commonalities and differences (also with respect to the effects on pain). Similarly, did the authors split trials according to concordant / discordant information in social and conditioning cues in order to assess what is happening when both types of information agree vs when they do not?

5. Finally, I find some phrases in the abstract to be slightly unfortunate. First, I do not really agree with their statement of “similar behavioral effects”, as i) social effects decreased over time, whereas conditioning effects increased over time, ii) social effects were significantly larger than conditioning effects, and iii) only the social cues affected SCR. To me that is a quite different picture. Second, the authors mention that their findings have important implications for theories of predictive coding, yet they do not elaborate on this at all when discussing their results.

Minor points

L315: The authors find a highly significant extinction effect on expectations for the social cues (reported in L137), so I am not sure why they speak about “non-extinguishing” effects of social information. What is interesting here though is that this effect does not transfer to pain ratings – could the authors comment on that?

L468: What was the criterion for excluding the two participants and how were those participants different from the rest of the sample?

L482: Could the authors state the reinforcement ratio (I assume it was 50%)?

L496: Were the participants debriefed at the end of the study and given the possibility to withdraw their data after they knew about the deception (i.e. the description of the ratings coming from previous participants)?

L558: I'm not sure I understand the sentence “Movement parameters were added as regressor of no interest to control for motion artifacts and spikes” – did the authors perform spike regression?

L561: It would be helpful to have a supplementary figure that explains the general mediation model and includes all path names (path c/c' is not present in any figure).

L577: The authors might want to motivate what exactly the word-cloud plots add to the analysis – to me this is not clear from the description given here.

L582: What is the rationale underlying this analysis and is this something that is routinely done in mediation analysis? What exactly would be the role of a brain region that shows such effects? I also do not understand what is meant by “ $P < 0.05$ FDR-corrected and adjacent voxels at $p < 0.05$ uncorrected” – do the authors indeed use uncorrected statistics to identify voxels for their conjunction? This would be a lower standard than what they apply in other analyses and make the results less convincing.

L874: There seems to be a quite strong block-wise effect on expectation ratings in Figure 1D. How do the authors explain this pattern?

L931: What is the activation threshold used here? What do the error-bands represent in the slope plot? There seems to be a high-leverage point in the right-most plot of Figure 3B: does the result hold with this point removed?

RESPONSE LETTER NCOMMS-18-37092

Reviewers' comments:

Reviewer #1 (Remarks to the Author):

The present study investigates how social influences and learning effects on pain are mediated in the human brain. 36 participants performed an fMRI experiment in which they provided pain single-trial pain expectation and pain intensity ratings. Prior to the pain stimuli fictive pain ratings of 10 other participants and a visual conditioning stimulus were presented. Mediation analysis was performed to identify brain areas mediating the social influence and conditioning effects on pain. The results showed social and conditioning effects on pain expectation and pain intensity ratings with the social effects being much stronger than the conditioning effects. The results further show that the brain mediators of both effects fundamentally differed. Social influences were mostly mediated by brain networks related to attention whereas conditioning effects were mostly mediated by the default mode network.

Understanding different contextual modulations of emotions and pain is a highly relevant and timely topic. The methods of the study are sound and timely. The results are novel and convincing. The interpretation is adequate. A few explanations and clarifications might help to further improve this excellent manuscript.

Response: We thank the reviewer for this positive evaluation of our manuscript.

1. The study includes a conjunction analysis of path a and path ab effects. This approach is novel and the underlying logic and its interpretation should be explained in plain terms.

Response: Thank you for this request to provide a clearer explanation of this analysis. We have now extended the explanation of the conjunction in the main text of the revised manuscript, p. 9:

“We used a conjunction analysis to identify spatial overlap in brain areas responding to the $Social_{HIGH} > Social_{LOW}$ manipulation (Path a) and those mediating the effects of this manipulation on pain outcomes (Path ab) (see Figure 3A and Methods). The mediation effect is driven by a combination of responses to the experimental manipulation, correlations with pain, and correlations in individual differences between these two effects (i.e., stronger responses to the $Social_{HIGH} > Social_{LOW}$ manipulation correlated with stronger effects of the brain region on pain). Thus, the ab effect encompasses all the elements required to link the manipulation, brain, and behavior, but does not guarantee that all significant regions show significant task effects. The conjunction analysis therefore shows regions that show both $Social_{HIGH} > Social_{LOW}$ effects and mediation effects. The conjunction confirmed the important contribution of the dlPFC, IPS, as well as dmPFC, vlPFC, and visual cortex to social information effects on pain. These areas showed increased activation for the $Social_{HIGH}$ compared to the $Social_{LOW}$ condition and this change in activation statistically explained the observed increase in pain rating ($Social_{HIGH} > Social_{LOW}$).”

And on page p. 10 (CS effects):

“We again used a conjunction analysis to illustrate the overlap in brain regions that showed increased activation to pain in the CS_{HIGH} compared to pain in the CS_{LOW} condition (path a effects), and regions that statistically mediated the effects of this experimental manipulation on pain ratings (path ab effects).”

Further, we have added an additional explanation of brain mediation analysis, the interpretation of the different paths, to the Suppl. Information (Suppl. Fig. S1, see below).

A) Effect of predictor (X) on outcome (Y)

B) Mediation of X-Y-Effect by mediating variable (M)

C) Brain mediation analysis (multi-level)

2. The participants provide pain intensity ratings. Other studies have shown that pain intensity and pain affect can dissociate. The authors might discuss the distinction between pain intensity and pain affect, how the social and learning effects might have influenced pain intensity and pain affect and how possible differences in the effects on pain intensity and pain affect might account for differences in the underlying brain mediators.

Response: This is a thoughtful and insightful comment, and we have added discussion of this possibility in the manuscript. In the present study, we were limited to one rating per trial, given the high number of trials we had and the significant amount of time (at least 5s plus potential jitter for each of 96 trials) a second rating would have required. While in general measures of pain intensity and pain affect are highly correlated (Wager, et al. 2004; Coghill et al. 1999), we agree that some studies have shown dissociations in effects on pain intensity and pain affect (e.g., Rainville et al. 1997, Villemure & Bushnell, 2009). Some interventions, including those of social touch and opioids in our prior work and others', find greater effects of psychological manipulations on pain affect than on pain intensity (e.g., Lopez-Sola, et al., 2018; Atlas, et al. 2012). Other studies have shown differential effects of different psychological manipulations (mood versus attention) on pain affect and intensity (Villemure & Bushnell, 2009). Thus, we agree that there might be different brain mechanisms involved in the representation of pain intensity and pain affect, and indeed, those may be differently affected by learning and social information. While such differences would still be reflected in the path *a* effects, they might not be fully captured in the path *ab* effects, since those relate to the behavioral outcome (pain intensity in this case).

Some features of our results do not fit with this theory, however. First, social information had stronger effects on pain intensity ratings than did conditioning, making it less plausible that social information primarily influences pain affect. Neither is conditioning likely to primarily influence pain affect; other studies, conditioning similar to that employed here has strong effects on pain intensity as well (e.g., Atlas et al. 2010; Koban, Kusko, Wager, 2018; Jepma et al. 2015, 2018). Third, the brain mediators of pain affect in other studies (Lopez-Sola et al. 2018, 2019) include areas that show strong effects social information and mediation of pain intensity here (e.g., vIPFC/OFC and dIPFC). Thus, though the differences in effects of social information and conditioning we observed here may be driven in part by differential effects on pain intensity vs. pain affect, we think other explanations are more likely.

We have now added these points to our revised discussion (page 20):

“Finally, we tested effects on ratings of pain intensity, but not pain affect (unpleasantness). These two measures are often highly correlated^{57,102}. Yet, pain intensity and unpleasantness

can be differentially modulated by contextual manipulations^{103,104} and those two aspects correlate with different brain areas^{105,106}. It is thus possible that some brain regions, such as the ACC, which showed increased activity due to our manipulation (i.e., path a effect), did not show strong mediation effects on intensity ratings (path ab), because they relate more to pain affect than to intensity. Future studies could test how social information and learning influence pain affect—a measure that is often more sensitive to social and contextual manipulations of pain¹⁰³.”

3. The general reader is likely interested in the number of participants and trials which should already been included in the results section.

Response: We have now included the number of participants and trials in the end of the introduction, where we provide an overview of the study design and procedures.

Page 4:

“In each of 96 trials of a learning task (see Figure 1A), 38 participants were presented with two types of information.”

4. The figure legends should include more information on the underlying statistics.

Response: We have now included more details on the statistics in the figure legends, and we also provide the underlying data in the source data file.

Fig. 1 caption (changes highlighted in yellow in the manuscript):

*“Effects on behavior, physiology, and a priori brain patterns. Violinplots show the distribution of individual effects (beta-estimates) for $Social_{HIGH} > Social_{LOW}$ and $CS_{HIGH} > CS_{LOW}$. Each dot reflects the beta estimate of one participant. Source data is available in the source data file. Both social information and CS significantly influenced expectation (social information: $t(35) = 10.65, p < 0.001$, CS: $t(35) = 3.02, p = 0.005$) and pain ratings (social: $t(35) = 6.19, p < 0.001$, CS: $t(35) = 3.13, p = 0.003$). Skin conductance responses (SCR) were significantly modulated by social information ($t(35) = 2.04, p = 0.049$) but not by CS ($t(35) = 1.47, p = 0.15$). Neither NPS nor SIIPS showed significant responses (as measured by the dot product) to social information or CS, suggesting that their effects on behavior must be mediated by other brain mechanisms. **D) Time course of expectation and pain ratings.** The difference between dotted and solid lines reflects the CS effect, and the difference between gray and black lines the social information effect. CS effects on expectation and pain increased over time (Interaction effects $CS*Time$ on expectation: $t(35) = 3.02, p = 0.004$, on pain: $t(35) = 2.03, p = 0.049$). Social information effects on expectations and pain remained significant throughout the experiment, but decreased over time for expectation ratings ($Social*Time$ on expectations: $t(35) = -4.57, p < 0.001$). The x-axis shows trials per condition. Source data with individual data for the time courses is available in the source data file. **E) Behavioral mediation analysis.** Expectation ratings significantly mediated both social ($t(35) = 8.56, p < 0.001$) and CS effects on pain ($t(35) = 3.27, p < 0.001$). Source data for panels C and D are provided as a Source Data file.”*

Fig. 2 caption now includes descriptive statistics for network similarities (see additions and edits in manuscript, highlighted in yellow).

Fig. 3 caption...: “Conjunction effects are displayed as the intersection of activation for paths a and ab, each of them thresholded at $P < 0.05$ FDR-corrected and adjacent voxels at $p < 0.05$ uncorrected. Shaded error bands reflect bootstrapped 95% confidence intervals.”

Fig. 4 caption (new figure): “*Difference between social and CS effects in frontoparietal subnetworks. A) Difference in mean Path a beta weight (Social – CS) in Control A, Control B, and Control C subnetworks in left and right hemisphere. Significantly greater activation for Social information was found in the Control A (left: $t(35) = 2.3$, $p = 0.027$, 95%-CI = [0.03, 0.39], Cohen’s $d = 0.38$; right: $t(35) = 2.9$, $p = 0.006$, 95%-CI = [0.08, 0.43], Cohen’s $d = 0.48$) and the Control B (left: $t(35) = 2.6$, $p = 0.012$, 95%-CI = [0.08, 0.59], Cohen’s $d = 0.44$; right: $t(35) = 2.6$, $p = 0.013$, 95%-CI = [0.07, 0.51], Cohen’s $d = 0.44$), but not in the Control C network. Stars denote networks with significant differences. Source data are provided as a Source Data file. B) Display of control subnetworks A, B, and C on sagittal and transversal brain slices.*”

Reviewer #2 (Remarks to the Author):

The study by Koban and colleagues takes an innovative approach to dissecting the effects of different sources of pain-altering expectations. The topic itself is of great interest and the authors address it from multiple angles: analysis of self-report and peripheral physiological data, mediation analysis of aforementioned data and fMRI data, correlation with large-scale brain networks and relating their effects to meta-analytic data. The manuscript is very well written and mostly easy to follow, though a few parts could be clarified. The sample size is adequate and is furthermore based on a behavioural study using a very similar paradigm. My main criticism (as detailed below) relates to the authors’ analytical strategy and the robustness / reproducibility / generalizability of the ‘social cue’ vs ‘conditioned cue’ effects due to peculiarities of the paradigm.

1. According to the last paragraph of the introduction, the authors’ main hypotheses are: “If associative learning and social information effects have common brain mechanisms, brain effects should be found in the same or similar brain regions. Alternatively, if the effects of conditioning and social information have separable underlying mechanisms, brain effects should be seen in distinct brain areas”. However, I struggle to find a formal statistical test in the paper that tests for commonalities or compares the evidence for one versus the other hypothesis. Commonalities could for example be investigated with a simple conjunction analysis (for path a, path ab or a combination of the two). Their SVM analysis (L299) does not really address this point, as it doesn’t exclude the possibility of shared activations (as the authors state themselves), but only shows that differentiation is possible. Could they maybe also test how much variance is shared between SOCIAL and CS effects in terms of loading on different large-scale networks (their Figure 2)?

Response: We thank the reviewer for the detailed reading of our manuscript and the constructive feedback. While we have focused our presentation on the differences, we agree that there may also be shared mechanisms and we have now added a direct investigation of those common brain areas, resulting in a more complete description of our results. We have added a conjunction analysis for social and CS path a and path ab effects to Figure 2. To avoid false negatives and to include commonalities at a less conservative level of significance, we extended the displayed activations to 0.05 uncorrected (adjacent to FDR-corrected peak activations, see modified Figure 2). At this level and at 0.01 uncorrected (adjacent to peak voxels at 0.05 FDR corrected), overlapping activity can be observed in a few frontal and parietal clusters (especially for the mediation effect, Path ab, see bottom panel D):

Further, we computed Dice coefficients to quantify the amount of overlap compared to non-overlap for those effects, and we have added the following paragraph to our results section, p. 10-11:

“Similarity versus separability of social influence and learning effects. To test for commonalities between social information and learning effects, we performed a conjunction analysis (using a conjunction-null⁴⁹). At a lenient threshold ($p < 0.05$ uncorrected voxels adjacent to FDR-significant voxels), CS and social influence showed a few small clusters of common Path a effects in visual cortex and bilateral superior parietal lobule (see Fig. 2C). To quantify the amount of shared versus separate activations, we computed the Dice coefficient for voxels activated at 0.05 uncorrected. The Dice coefficient—which can theoretically range between 0 and 1, where 0 reflects complete separation and 1 reflects perfect overlap—for Path a was 0.024, thus suggesting relatively small overlap between Path a activations for Social and CS effects. For path ab, shared mediation effects of social influence and CS effects on pain were observed at $p < 0.01$ and $p < 0.05$ uncorrected, notably in dmPFC, dlPFC, IFG, and IPL, with a Dice coefficient of 0.081. Thus, while these clusters of conjunction effects were relatively small in size, and mainly found in adjacent to the peak effects for each condition, this tentatively suggests that some parts of the frontoparietal network may be involved in top-down modulation of pain based on expectations irrespective of their sources.”

The dice coefficient is similar to spatial correlation for binary (thresholded) images. Regarding shared and unique variance in different networks, please see also our response to point #3 (networks).

2. With regards to the effects of social and conditioned cues, it seems to me as if the paradigm design might prevent any strong effects of conditioned cues on pain. First, there is a 50% reinforcement ratio,

which is quite low and will lead to slow acquisition of contingencies (which is compounded by presentation of the social cues at the same time). Second, the choice of using trace conditioning will also play a role, as the delay between cue and pain is longer than the interval between the end of the current trial and the start of the next – such timing is known to weaken associative learning. At the same time, the peculiarities of the design might actually boost the social effects: since participants were explicitly instructed to detect the CS-US contingencies, their attention will have been focused on the CS cues and one might wonder whether they had enough cognitive capacity to detect to non-informativeness of the social cues with regards to the level of pain. So I am not sure the large social effect will be easily replicated in other experiments once certain parameters of the design are changed.

Response: These are astute comments, and we agree that the sizes of the effects (both conditioning and social) will be influenced by multiple parameters, including CS-US delay (here, trace conditioning) and the presence of the compound cues, as the reviewer suggests. We do have evidence, however, that 1) trace conditioning produces reliable CS effects on pain and that 2) the social effects are large and robust even without the compound cues.

Regarding point 1), multiple studies from our lab and other groups have used trace conditioning paradigms and shown robust effects of CS-high versus CS-low on pain (e.g., Atlas, et al., 2010; 2012; Koban, Kusko, Wager, 2017; Sharvit, et al., 2015). Of note, many studies, using trace (Atlas, et al., 2010; 2012; Schafer et al., 2015) or delay conditioning (Colloca, et al., 2008; 2010) actually use instructions in combination with conditioning. When no instructions are used, some participants do not learn the CS-contingency, even in the absence of social information (see Koban, Kusko, Wager, 2017).

Regarding point 2), we have replicated the social influence effect in multiple datasets and stimulus domains, including with separate social and conditioning tasks (i.e., simple rather than compound cues), which showed large social influence effects (see below). In addition, other studies, e.g., by Willroth, Koban, Hilimire (2017) and Yoshida et al. (2013) have shown social information effects without compound cues. Willroth et al. (2017) showed robust effects of social influence on affective picture processing both in terms of behavior and EEG responses, in the absence of potentially distracting learning cues. We have also recently acquired (unpublished) data on social influences (in the absence of conditioning) on other affective or value-related stimuli such as empathy-for-pain and food wanting, and we find robust social influence effects in all those domains. The figure below shows unpublished behavioral data from a pharmacological study (Koban, Kusko, Lopez-Sola, & Wager, in preparation) in which participants separately performed a social influence task and a CS task (order counterbalanced across participants). This study further differs from the present fMRI study in the length of the pain stimulation (plateau duration of 4s, using an ATS instead of a CHEPS thermode). Plots below show the results of this study (no-drug condition only, N=25) and indicate very strong social influence and CS effects in these tasks, even if the two types of information are not presented in the same task. If the editor and reviewers wish, we could add this figure and a description of the behavioral study to the supplementary information. We believe this may not be necessary to make the central points of the present manuscript, however, as it concerns the generalizability of the effects across different paradigms than those we focused on here.

In sum, the both the CS and the social influence effects are robust to variations in the design and even across different modalities (pain, affective images, food cues). Given the strength of

the social information effects, we completely agree with the reviewer that future studies should investigate how social information may prevent or reduce CS learning (since the CS effects may indeed be smaller in this type of design than if presented in isolation). **We have emphasized this point in our revised discussion, p. 14:**

“Future studies should test whether the presence of salient social information can hinder experience-based learning, by comparing CS effects in the presence versus absence of salient social cues.”

We have also added the following to our discussion regarding trace and delay conditioning and the possible interactions with social information effects (p. 19):

“Future studies could also test whether trace and delay conditioning interact in different ways with social information and instructions. For instance, it is possible that the effects of the social information are stronger when participants see both type of cues at the same time, since they may be too distracted to detect that the social information was not actually predictive. We note, however, that this explanation is unlikely, since several other studies have shown robust effects of social information on affect ratings and brain responses in the absence of conditioning cues^{62,93}”

And to the conclusions, p. 21:

“Future studies should test how social influence may enhance or prevent experience-based learning.”

3. I am also not sure whether the differential mapping on large-scale networks will be a robust phenomenon. For example, the way the social information was presented might automatically involve fronto-parietal systems due to the complexity of the input and the need to average the ratings to form a mean (this might be replayed during the pain stimulation). Would one expect to see a similar effect if the social information had instead been presented with strongly reduced complexity, e.g. as one number reflecting the mean rating, which would need much fewer cognitive resources? I also wonder about the level of granularity chosen here for the large-scale networks: is seven networks the optimal level for looking at such specific effects? Might it not be helpful instead to investigate this (very interesting) question across different levels of granularity in order to check whether similar effects of separable networks are observed? In my eyes, this would clearly strengthen the paper.

Response: Regarding the first point here, we understand how integrating information about multiple other individuals' ratings (lines on the display) could be seen as more cognitively demanding than other, simpler cues. However, several considerations mitigate this to some degree. The visual information presented on the VAS corresponds to visual representation of 'stars' and other typical displays in online ratings. It also closely maps to the participants' own ratings on the VAS. Given the strong effect of this type of information in multiple datasets (see above, Koban & Wager, 2016; Willroth, et al., 2017; Koban, et al., unpublished), we think that participants understand them quite intuitively. In addition, other studies of cue-induced pain modulation using simple shape cues have also produced effects on pain that are mediated in part via the dorsolateral prefrontal cortex (dlPFC; Atlas et al. 2010; Jepma et al. 2018), so we don't think that dlPFC engagement necessarily results from cognitive load per se. **We agree, however that future studies could productively compare different formats of presentation (see Lobanov et al., 2014) and different levels of cognitive demand (or effort) and we have added this point to the limitations section in the discussion, p. 19:**

“[...], future studies may use different cue types (e.g., using numerical instead of visual representations of others' ratings) and other modalities of cue presentation⁶⁸ to replicate the present findings.”

Regarding the networks, we chose the seven-network solution as a standard, a priori test, since these seven networks have established (although debated) functional associations. We agree, however, that showing network overlap at a finer level of granularity would be an interesting comparison. An open question for future research or a methods paper is which degree of granularity and which parcellation are optimal for interpreting these and other effects.

To add different levels of granularity to the present set of findings, we have now performed several additional analyses (also addressing point #1).

First, we used a more fine-grained (16-network) parcellation from the same group (Yeo/Buckner), and we map individual Path a responses to subcomponents of the frontoparietal network (new Figure 4, parcellation and other networks are shown in Suppl Fig. S10). Interestingly, both Control networks A and B bilaterally, but not Control network C showed greater Social compared to CS effects. See additional results on p. 11-12 and new Figure 4:

P. 11-12: “[...], since conjunction effects were found in frontal and parietal regions, we tested how individual (unthresholded and normalized) beta images for Social and CS Path a effects engaged more fine-grained parcellations of the frontoparietal network, based on an established 16-network cortical parcellation^{47,50}. Overall, the frontoparietal network (but no other network) was significantly more activated for Social compared to CS Path a effects ($t(35) = 3.1$, $p = 0.0042$, Cohen’s $d = 0.51$). Further, the Control A and Control B subnetworks in both left and right hemispheres were more activated for the Social compared to the CS Path a effect (see Figure 4 for details). In contrast, both left and right Control C subnetwork did not differ between Social and CS effects. This pattern of findings is consistent with the observation that strong effects for social effects were observed in prefrontal, lateral parietal, and temporal parts of the frontoparietal network, but not in medial parietal cortex or posterior cingulate cortex (which constitute the Control C subnetwork).”

Suppl. Fig. S10:

Suppl. Fig. S10. Path a (for Social and CS) and Path b average beta weights in 16 bilateral (32) networks. Top row illustrates the brain parcellation used (Schaefer et al., 2018). Wedge plots depict mean beta values across voxels for each parcel (R, L = right, left hemisphere). Red wedges indicate

positive and blue wedges indicate negative values. Darker areas indicate SEM across individuals. Outer ring colors match the color coding of the brain display and the network names shown on the right. Source data are provided as a Source Data file.

Second, we investigated the spatial pattern of the mediation maps across all voxels within each of the seven main networks (Suppl. Figure S11). This analysis was inspired by population coding analyses in neuroscience that reveal overlapping populations of neurons within a region that have different functional properties. Here, voxels are the unit of analysis, and voxels within a network may have diverse functional properties in terms of relationships with CS and social cue effects. For each of the 7 main large-scale networks, we characterized the joint distribution of voxels that mediated social cues and CS effects. While network averages are often a useful summary of effects, this analysis acknowledges that there may also be substantial neural diversity within a network.

“[...] we analyzed the spatial covariation between the unthresholded weight maps for social and CS mediation effects (Path ab). [...] In Suppl. Fig 11, we plotted the weight (effect magnitude) of each voxel for the social mediation effect on the x- and for the cue mediation effects on the y-axis, separately for each network^{51,52}. Effects in any given voxel could be positive, negative, or near-zero for each of the Social cue and CS mediation effects. This lends itself naturally to classifying voxels within each network into 8 equally-sized octants depending on the relative Social cue and CS effects. Voxels in Octants 1 and 3 were selectively related to CS and Social cue mediation respectively, with positive effects. Octants 5 and 7 showed selective negative effects of CS or Social cues, respectively. Voxels in Octants 2 and 6 show positive and negative common activation for both cue types, respectively, and are thus those with common effects of both cue types in the same direction. Voxels in these octants drive positive spatial correlations across voxels within the network as a whole, indicating overlap. Finally, voxels in Octants 8 and 4 are those with positive CS effects but negative Social cue effects, or vice versa. Voxels in these octants drive negative spatial correlations, indicating dissimilarity. Furthermore, to provide an overall measure for voxels in each octant, we computed the sum of squared distances (SSD) from the origin, thus combining a measure of both absolute numbers of voxels in each octant and their (squared) distance from the origin (thus integrating beta weights for social and CS mediation).

Suppl. Figure S11 provides an overview of voxel weight distributions in each network. As can be seen, the distribution of Social and CS effects shows qualitatively different patterns across networks. Visual, ventral attention, and default mode networks have peaks in Octant 2, reflecting a disproportionately large number of voxels that show positive (if not necessarily significant) effects for both Social and CS mediation. This indicates overlap in some voxels in these networks. However, these networks also contained many voxels with positive weights only for CS (Octant 1) or only for Social mediation effects (Octant 3). Frontoparietal and dorsal attention networks show large effects in the shared positive (Octant 2) and in uniquely Social mediation effects (Octant 3). The limbic network shows a peak in voxels mediating CS but not (or even suppressing) Social effects (Octant 1 and 8). Finally, the Somatomotor network was associated with many voxels showing suppressor effects for CS (negative slopes in mediation) and shared suppressor effects for CS and social effects, in line with its role in primary nociceptive, but less in contextual pain modulation effects. Overall, this analysis provides more detailed evidence for shared and non-shared elements within each network. The networks with the strongest evidence for some shared processing include the Dorsal and Ventral Attention, Frontoparietal, and Default Mode networks, but these similarities are offset by the differential responses in the vast majority of voxels in these networks (the overall

spatial covariance across voxels is relatively weak). Those with the strongest evidence for dissimilar effects of Social and CS cues include the Limbic and Somatomotor networks.”

Suppl. Fig. 11

Suppl. Fig. S11. Display of spatial covariation of mediation maps for Social and CS effects on pain. A) Scatter plots, displaying unthresholded single-voxel beta weights for CS and Social mediation effects within each of seven resting state-networks (from top to bottom: Ventral Attention, Limbic, Default, Frontoparietal, Dorsal Attention, Somatomotor, and Visual network). Different colors are assigned to eight octants, that reflect voxels showing positive mediation for CS but not social effects (octant 1), positive mediation effects for both (octant 2), positive mediation of social, but not CS effects (octant 3), and so on. Radial grids display distance from the origin (0,0) in 0.02 steps. B) Bars show sum of squared distances from the origin for all network voxels per octant, thus reflecting both number of voxels in each quadrant and their quadratically scaled beta weights. Source data are provided as a Source Data file C) Surface displays of networks (in bright gray), overlaid with unthresholded voxels for the 1-3 octants, indicating weights positively modulated by CS (octant 1), by social information (octant 3), or by both (octant 2). Note that those maps are purely descriptive and unthresholded illustrations of mediation beta weights.

We further discuss the idea of comparing different types of parcellations and networks as an important future direction (edits on page 20):

“Further, we tested the mapping of our brain results on a 7-network parcellation of resting-state activity⁴⁷, since these networks are relatively well-known and established. We complemented this approach by investigating the contribution of frontoparietal subnetworks and 16 smaller unique networks in each hemisphere. We further characterized the distribution of Social and CS mediation effects across individual voxels within those seven networks. There are many ways, however, to characterize and summarize these effects and their similarities and differences. It is thus an open question for future studies to compare different cortical (and subcortical) parcellations to find the optimal degree of granularity for characterizing the functional patterns underlying different pain-modulatory effects.”

4. There are also some analyses that could add valuable information to the paper. The authors aim to dissociate the effects of social cues and conditioned cues on pain, but they do not report any data on the effects the cues had on peripheral physiological and brain data at the time of cue presentation. I would urge the authors to do so, as the cue response might be interesting with regards to commonalities and differences (also with respect to the effects on pain). Similarly, did the authors split trials according to concordant / discordant information in social and conditioning cues in order to assess what is happening when both types of information agree vs when they do not?

Response:

Thank you for those helpful suggestions. Regarding the peripheral physiological responses to cues: Similar to our behavioral study (Koban & Wager, 2016), we did not observe any clear SCR at the time of cue presentation (see left panel below). We have now added these time-locked average SCR curves for both cue and 49°C-pain onset by condition, as well as pain-related responses by temperature, to the supplementary materials (Suppl. Fig. S2):

We have now performed additional brain mediation models to investigate the effects of social and CS information on brain activity at the time of cue presentation and their effects on expectation ratings. While these effects were less strong than the effects during pain (i.e., for CS, they mostly do not survive FDR correction), we agree that they might still be informative. We have added a short description to the results (see further below) and a figure to the Supplementary Materials (Suppl. Fig. S12).

A) CUES → BRAIN → EXPECTATION multi-level mediation model

B) Path b: Brain activity correlating with trial-by-trial variation in expectation ratings, independent of CS and social information

C) Path a: Cue-related activity for Social_{HIGH} > Social_{LOW} and for CS_{HIGH} > CS_{LOW}

D) Path a: Mediation effects for Social and for CS effects on expectations

Regarding concordant/discordant effects: We agree that interaction effects (which would test the effects of concordant versus discordant information) would be an interesting additional question. However, during the pain stimulation, paralleling the behavioral effects (which did not show any CS*Social interaction), we did not see any strong or meaningful effects for the interaction between CS and Social information during pain, even at very liberal thresholds (0.05 uncorrected):

This suggests that the effects of the social and conditioned cues were mainly additive. We have added a statement to that effect in the manuscript on p. 13-14 (see below). We do see, however, some interaction effects (at uncorrected levels) during the presentation of the cues,

which we have now added to the supplementary materials (Suppl. Fig. S13) and in the manuscript (p. 14):

We added the following paragraph to the revised results section, page 13-14:

“Brain activity at the time of cue-presentation and interactions between social and learning effects. For completeness, additional mediation models tested how CS and social information influenced brain activity at the time of cue presentation, and how cue-related brain activity mediated effects on expectations, again demonstrating largely different systems responding to Social and CS cues (see Suppl. Fig. S12). The effects during cue processing differed from the effects observed during pain. At uncorrected thresholds ($p < .001$), increased activity to CS_{HIGH} compared to CS_{LOW} cues was found in vmPFC, striatum, and superior frontal gyrus, whereas increased activity to $Social_{HIGH}$ versus $Social_{LOW}$ cues was found in anterior insula, amygdala, hippocampus, thalamus, brainstem, occipital, and medial frontal and parietal areas. Small areas of overlapping social and CS Path a effects were observed in the vmPFC, striatum, and the brainstem. Hippocampus and medial prefrontal areas mediated CS effects on expectations, while a small cluster in the right dlPFC, in the left postcentral gyrus, and several occipital areas mediated social cue effects on expectations. We further tested for interactions between social and CS conditions. At an uncorrected level, interaction effects, suggesting greater cue-responses to congruent than incongruent social and CS cues, were found in several areas, including medial and inferior temporal lobe (including hippocampus), putamen, and ventral parts of the insula (displayed in Supplementary Figure S13). No meaningful interaction effects were found at the time of pain stimulation, in line with the additive effects of CS and social information on pain reports.”

5. Finally, I find some phrases in the abstract to be slightly unfortunate. First, I do not really agree with their statement of “similar behavioral effects”, as i) social effects decreased over time, whereas conditioning effects increased over time, ii) social effects were significantly larger than conditioning effects, and iii) only the social cues affected SCR. To me that is a quite different picture. Second, the authors mention that their findings have important implications for theories of predictive coding, yet they do not elaborate on this at all when discussing their results.

Response: We agree that there are differences as well as similarities in the behavioral effects of social information and conditioning. We have now made edits to the abstract to better reflect both similarities and differences and clarify our intended message. In pointing out the similarities, we wanted to draw attention to the fact that both types of information have significant and reproducible effects on pain, which are mediated by consciously accessible expectations. In our previous behavioral and psychophysiological study, we further showed robust effects on SCR for both types of information, which seem to be less detectible here due to scanner noise. Thus, from looking at behavior *alone*, one could assume that expectations may be a common mechanism for both effects and thus that the brain mechanisms underlying

them would also be similar. However, we found that two effects that are both mediated by expectancies can be shaped based on different neurophysiological pathways.

The abstract now reads:

“[...] Social information and conditioned stimuli had significant effects on pain ratings, and both effects were mediated by self-reported expectations. Yet, at the brain level, these effects were mediated by largely separable brain patterns, which mapped onto different large-scale functional networks.”

We have also added more discussion regarding predictive coding to the revised discussion, p. 14-15:

“These findings have implications for predictive coding theories of pain and of information processing more broadly². Whereas previous studies have established the effects of expectations on information processing in many domains, it is less clear where these expectations come from and which brain systems are at the ‘top’ of ‘top-down’ modulation of information processing⁵⁶. Our results suggest a distributed system of source-dependent (e.g., social or conditioned cues) and source-independent brain areas involved in modulating pain based on expectations. Current predictive coding models focus on the functional form of the models (e.g., hierarchical Bayesian information) but not potential differences across priors. We show that priors in predictive coding frameworks may arise from different neural sources, even if they have common psychological correlates.”

And in the conclusions, p. 20-21: *“Together, these results suggest that top-down modulation of experience can stem from distributed sources in frontoparietal and default mode areas, depending on the source of information.”*

Minor points

L315: The authors find a highly significant extinction effect on expectations for the social cues (reported in L137), so I am not sure why they speak about “non-extinguishing” effects of social information. What is interesting here though is that this effect does not transfer to pain ratings – could the authors comment on that?

Response: We would like to clarify that the social cue effect on the pain ratings does not change over time, whereas the effect on expectation ratings becomes somewhat smaller, but also remains large and significant throughout the whole experiment. While we did not have an *a priori* hypothesis regarding this dissociation, one plausible interpretation is that participants' expectations at the very beginning of the experiment are purely driven by the social information, and later also integrated with learned information about the general level of pain to be expected in this experiment. Anecdotally, at the time of the debriefing, participants would for example comment on that they were overall less (or more) pain sensitive than the people whose pain ratings they observed, so a reasonable explanation is that people learn to take these additional factors into account when rating expected pain, and calibrate what the social cues mean for them based on their experience in the study.

L468: What was the criterion for excluding the two participants and how were those participants different from the rest of the sample?

Response: Both of these participants had multiple issues. First, both of these two subjects arrived late and/or caused further delays during the scanning that made the completion of all runs of the subsequent generalization task (to be reported elsewhere) impossible. Second, both subjects had significant head movement visible to the scanner techs during acquisition, and one of them also had visible movement-related artifacts in the T1-weighted image. The presence of large movements (>3mm total) was later confirmed during preprocessing. Thus,

we decided before study completion and data analysis to exclude these two subjects and replaced them with two additional subjects in order to keep the assignment of learning cues completely counterbalanced (since we had 18 different counterbalancing conditions).

L482: Could the authors state the reinforcement ratio (I assume it was 50%)?

Response: Yes, 50% of the CS-high cues were followed by 49°C and 50% by 50°C stimulation. 50% of CS-low cues were followed by 48°C and 50% by 49°C stimulation. We have now included this information more prominently already in the introduction on page 4:
"[...] low-to-medium (CS_{LOW} , 50% 48°C and 50% 49°C) or medium-to-high (CS_{HIGH} , 50% 49°C and 50% 50°C) thermal pain stimulation."

L496: Were the participants debriefed at the end of the study and given the possibility to withdraw their data after they knew about the deception (i.e. the description of the ratings coming from previous participants)?

Response: Participants were fully debriefed at the end of the study. Participants were told in writing and in person and thus knew that they were free to withdraw their participation at any time, but none chose to do so. Since this experiment involved only minimal deception, in the sense that those ratings were similar enough to actual other people's ratings (which vary widely for those stimuli) (see Yoshida, et al., 2013; Koban et al., 2016 for parallel procedures), participants were not given an explicit choice at the end of the study to withdraw their data. As stated in the manuscript, this procedure was approved by the IRB of the Department of Psychology and Neuroscience at the University of Colorado Boulder.

L558: I'm not sure I understand the sentence "Movement parameters were added as regressor of no interest to control for motion artifacts and spikes" – did the authors perform spike regression?

Response: We have now clarified this in the methods. A standard preprocessing procedure in our lab (e.g., Atlas et al., 2010, 2012; Woo et al. 2015, 2017; Koban, et al., 2017; Jepma, et al., 2018; and others) is to identify time points that are potential outliers (or 'spikes') based on meeting any of the following criteria: (a) absolute value of global signal > 10 median absolute deviations (m.a.d.); or (b) mahalanobis distance across slice-specific global means and spatial standard deviations > 10 m.a.d. These time points are identified on a per-run basis using recursive exclusion of outliers in a step-down test, so that outliers are removed before recursively identifying additional outliers (3 iterations). Any time points identified in this way are modeled during first-level regression analysis with an individual dummy regressor containing a value of 1 for the outlier time point and zero for all other time points. This procedure effectively removes influences of these time points without disrupting the autocorrelation structure in the time series and with degrees of freedom properly accounted for in the first-level model. For typical participants across studies, this procedure identifies 1-3% of time points as outliers and is thus more conservative than many currently used 'scrubbing' procedures, with minimal impact on data missingness. Our participants here were typical in terms of the numbers of time points excluded. In addition to these 'spike' regressors, our standard practice has been to add additional nuisance covariates to first-level models, including: (a) movement estimates for each run (6 parameters, 3 displacement and 3 rotation, each in 3 dimensions), (b) derivatives of movement estimates, (c) squared movement estimates, and (d) derivatives of squared movement estimates, for a total of 24 additional movement-related regressors per run (see Lund 2006). Variance inflation factors (VIFs) were examined to test for multicollinearity with regressors of interest, and trials with VIFs exceeding 2.5 were excluded from the subsequent mediation analysis (less than 5% of trials). **We have now expanded our description as following:**

Page 23: "Prior to preprocessing of functional data, time points that are potential global outliers (spikes) were identified based on meeting any of the following criteria: a) absolute value of global signal > 10 median absolute deviations (m.a.d.), or b) mahalanobis distance across slice-specific global means and spatial standard deviation > 10 m.a.d.. These time

points are identified on a per-run basis using recursive exclusion of outliers in a step-down test, so that outliers are removed before recursively identifying additional outliers (3 iterations).”

Page 24: “A total of 24 movement regressors (movement estimates for displacement and rotation in three dimensions, their derivatives, squared movement estimates, and derivatives of squared movement estimates) and spike regressors (coded as 1 for the outlier time point and zero for all other time points) were added as regressor of no interest to control for motion artifacts. Single trial regressors that had variance inflation factors > 2.5 (indicating potential multicollinearity) were excluded from subsequent analyses.”

L561: It would be helpful to have a supplementary figure that explains the general mediation model and includes all path names (path c/c' is not present in any figure).

Response: We thank the reviewer for this suggestion. We have now added a supplementary figure and text (see below) that explains the meaning of these paths in more detail (since Path c is the direct path without accounting for the mediation path, it is not possible to show it on the mediation model figure itself).

Suppl. Fig. S1:

“Suppl. Figure S1. A general overview of mediation analysis. **A)** The starting point of a mediation analysis is a directional relationship between an initial variable (X) and an outcome or dependent variable (Y). This effect is the Path c, the direct path from X to Y. Often, as in this case, X is an experimental manipulation, which provides stronger justification for inferring causal effects. **B)** Mediation tests whether a third variable (the mediator, M) can explain the effect of X on Y. Mediation assumes that X causes a change in M (Path a), which in turn leads to changes in Y (Path b, controlling for Path a effects). For instance, a ‘HIGH’ cue (compared to a LOW cue, X) may change expectations (M), which in turn change experienced pain (Y). The mediation effect (Path ab) is significant if it reduces the variance explained by the direct path (Path c’ – Path c). **C)** In a multilevel brain mediation analysis, individual single-trial estimates of brain activity can be used to test for mediation effects. In a mediation effect parametric mapping analysis (Wager et al., 2008), the effects shown in panel B) are

tested for every voxel in the brain (or in a subset of ROIs). Alternatively, multivariate pattern expression or average region of interest (ROI) activity can be tested as a mediating variable. Mediation effects are tested both on the single-person-, and across single-person- and group-level.”

L577: The authors might want to motivate what exactly the word-cloud plots add to the analysis – to me this is not clear from the description given here.

Response: We have added these Neurosynth decoding results as an additional way (in addition to univariate mediation findings and network similarity) to contextualize the brain results and provide a more fine-grained analysis of the functions associated with the activations in these areas (since the 7-network solution as the reviewer pointed out above, is somewhat coarse).

The Path *b* effects, for instance, load strongly on the somatomotor network—which is related to many different motor and somatosensory functions, and would also result from a finger tapping task or a pleasant somatosensory stimulation. The Neurosynth decoding analysis and the resulting word cloud plots provide an additional, more fine-grained contextualization of these results, namely pointing at the painfulness of the stimulation ('heat', 'pain', 'noxious', etc.) and the location of the stimulation ('limb', 'foot'). While we have pointed out in the text that these decoding results are only suggestive and should be interpreted with caution, we think they add an interesting additional level of inference to our manuscript. In particular, they identify the topics most consistently identified in other papers that show similar patterns of activation.

We have now clarified this motivation in our revision (p. 7):

“While relationships with terms should be interpreted with caution and are only suggestive, they can be useful for contextualizing findings and provide a more fine-grained comparison to existing large-scale databases than the canonical seven-network parcellation.”

L582: What is the rationale underlying this analysis and is this something that is routinely done in mediation analysis? What exactly would be the role of a brain region that shows such effects? I also do not understand what is meant by “ $P < 0.05$ FDR-corrected and adjacent voxels at $p < 0.05$ uncorrected” – do the authors indeed use uncorrected statistics to identify voxels for their conjunction? This would be a lower standard than what they apply in other analyses and make the results less convincing.

Response: Since Figure 2 shows a large number of brain regions, our rationale for the additional conjunction analysis was to show a clearer picture of the areas that are key for social and learning effects and consistently show both 1) increased activity to the respective HIGH > LOW conditions and 2) mediation of experimental effects on behavioral outcomes, thus explaining variance in the changes in pain intensity ratings. This test uses the conjunction null (Nichols, et al., 2005), which requires that each effect be significant. The reviewer is correct that we visualized results from the conjunction analysis that were significant at an uncorrected level, but only if those were adjacent to voxels exhibiting effects at the more stringent FDR corrected level. This is a reasonably common practice and serves to illustrate the extent of activation around a corrected peak, while still clearly demarcating regions significant at the corrected level. We now clarify:

Page 9: *“We next used a conjunction analysis to identify spatial overlap in brain areas responding to the $Social_{HIGH} > Social_{LOW}$ manipulation (Path *a*) and those mediating the effects of this manipulation on pain outcomes (Path *ab*) (see Figure 3A and Methods). The mediation effect is driven by a combination of responses to the experimental manipulation, correlations with pain, and correlations in individual differences between these two effects (i.e., stronger responses to the $Social_{HIGH} > Social_{LOW}$ manipulation correlated with stronger effects of the brain region on pain). Thus, the *ab* effect encompasses all the elements required to link the manipulation, brain, and behavior, but does not guarantee that all significant regions show significant task effects. The conjunction analysis shows therefore shows regions that show both $Social_{HIGH} > Social_{LOW}$ effects and mediation effects. [...].”*

We further extend our description in the methods, page 25:

“To test for overlap between Social Information and CS effects, we computed the intersection of brain areas that showed effects for both type of cues at $P < 0.05$ FDR-corrected and adjacent voxels at $p < 0.05$ uncorrected, for both Path a and Path ab effects (Figure 2). Further, to test for brain areas that showed both mediation (Path ab) and Path a effects, we computed the intersection of brain areas that showed both Path a and Path ab effects at $P < 0.05$ FDR-corrected and adjacent voxels at $p < 0.05$ uncorrected, separately for Social Information (Fig. 3A) and CS effects (Fig. 3B). This test uses the conjunction null⁴⁹, which requires that each effect be significant. We visualized results from the conjunction analysis that were significant at an uncorrected level, but only if those were adjacent to voxels exhibiting effects at the more stringent FDR corrected level.”

L874: There seems to be a quite strong block-wise effect on expectation ratings in Figure 1D. How do the authors explain this pattern?

Response: We agree that there is a slight increase in pain over time, as well as a block-wise fluctuations in expectation ratings (and potentially pain). Influences of temporal order on pain are often observed in our lab and are consistent with the findings and the computational model of Jepma, Jones, & Wager, (2014 Journal of Pain), demonstrating skin-site specific habituation to repeated pain, but skin-site independent (thus, presumably central) sensitization to repeated heat pain stimulation. In order to reduce the impact of the first heat stimulation on a new skin site, we usually present a short 49°C before the beginning of a new block and run. This seem to have reduced the impact on pain rating, but there seems to be still some effects on expectation ratings (which we assume reflect that participants may be more wary at the beginning of a new skin site). We have now added this in more detail on p. 17 and reference the abovementioned paper.

P.17: *“In order to habituate the participants to the stimulation on a new skin site¹⁰⁷, each run was preceded by a 49°C stimulus (1s plateau) before the scanner started again⁴¹.”*

L931: What is the activation threshold used here? What do the error-bands represent in the slope plot? There seems to be a high-leverage point in the right-most plot of Figure 3B: does the result hold with this point removed?

Response: We have now added the activation threshold to the figure legend (see also two points above). The error-bands reflect bootstrapped 95%-confidence intervals. We now added this information to the revised figure caption.

Fig. 3 caption:

“Conjunction effects are displayed as the intersection of activation for paths a and ab, each of them thresholded at $P < 0.05$ FDR-corrected and adjacent voxels at $p < 0.05$ uncorrected. Shaded error bands reflect bootstrapped 95% confidence intervals.”

We used robust regression to estimate the group effects, which is designed to minimize the influence of potential outliers without requiring ad hoc, case-by-case analyst decisions; but in response to this comment, we have now also re-run the analysis without the high-leverage point on the right. While the correlation is still positive, it does not reach significance anymore. However, we think this plot is still useful to illustrate individual variation in the two effects. While this point may be considered an outlier on statistical grounds (it is approximately 3 STD above the mean for each of the two variables), we think it is not caused by noise but by meaningful variation in individual learning. Thus, if the reviewers and editors agree, we would add the following statement to the results, in order to clarify that the point on the right might be driving the results:

“This association was modulated by individual differences in how much expectations were driven by the learning cues ($\beta = 0.21 [0.07, 0.35]$, $t(34) = 2.84$, $p = 0.008$, Cohen’s $d = 0.49$; Figure 3B, right)—yet this effect with individual differences was largely driven by a single participant (~ 3 STD above the mean on both brain and behavior, top right on the scatter plot). Larger sample sizes are needed to investigate individual differences in more detail.”

Otherwise, if the editor and reviewers prefer, we could also remove the two scatterplots related to individual differences from this figure.

REVIEWERS' COMMENTS:

Reviewer #1 (Remarks to the Author):

This reviewer's comments have been convincingly addressed. Congratulations on an excellent paper.

Reviewer #2 (Remarks to the Author):

The authors have comprehensively addressed all the points I raised and I think that the manuscript has substantially improved by the additional analyses the authors included. The breadth of novel analysis they now report is impressive and paints a more nuanced picture - I enjoyed reading this revised version of their manuscript a lot.

The only minor improvements I would suggest relate to Figure S11. Here, the coloring scheme gives a lot of weight to octants 1-3 and it is hard to properly judge the responses in the remaining octants as they have more shading / transparency. Could the authors also maybe add a legend to the plot indicating what each octant signifies in terms of shared / unique responses and direction of response?